# A RanGTP-independent mechanism allows ribosomal protein nuclear import for ribosome assembly

**Sabina Schütz[1,2], Ute Fischer[1], Martin Altvater[1,2], Purnima Nerurkar[1,2], Cohue Peña[1], Michaela Gerber[1], Yiming Chang[1], Stefanie Caesar[3], Olga T Schubert[4,5], Gabriel Schlenstedt[3], Vikram G Panse[1]\***

[1]Institute of Biochemistry, Department of Biology, ETH Zurich, Zurich, Switzerland; [2]Molecular Life Science Graduate School, University of Zurich, Zurich, Switzerland; [3]Institute of Medical Biochemistry and Molecular Biology, Universität des Saarlandes, Homburg, Germany; [4]Institute of Molecular Systems Biology, Department of Biology, ETH Zurich, Zurich, Switzerland; [5]Systems Biology Graduate School, Zurich, Zurich, Switzerland

**Abstract** Within a single generation time a growing yeast cell imports ~14 million ribosomal proteins (r-proteins) into the nucleus for ribosome production. After import, it is unclear how these intrinsically unstable and aggregation-prone proteins are targeted to the ribosome assembly site in the nucleolus. Here, we report the discovery of a conserved nuclear carrier Tsr2 that coordinates transfer of the r-protein eS26 to the earliest assembling pre-ribosome, the 90S. In vitro studies revealed that Tsr2 efficiently dissociates importin:eS26 complexes via an atypical RanGTP-independent mechanism that terminates the import process. Subsequently, Tsr2 binds the released eS26, shields it from proteolysis, and ensures its safe delivery to the 90S pre-ribosome. We anticipate similar carriers—termed here escortins—to securely connect the nuclear import machinery with pathways that deposit r-proteins onto developing pre-ribosomal particles.

**\*For correspondence:** vikram.panse@bc.biol.ethz.ch

**Competing interests:** The authors declare that no competing interests exist.

**Reviewing editor**: Ramanujan S Hegde, MRC Laboratory of Molecular Biology, United Kingdom

## Introduction

Ribosome assembly is an essential process that is tightly connected to cellular growth and proliferation (*Warner, 1999*). In the eukaryotic model organism budding yeast, this universal translating machine is built of two subunits: a large subunit (60S) consisting of three different rRNAs (25S, 5.8S, 5S) and 46 ribosomal proteins (r-proteins) and a small subunit (40S) that contains a single rRNA (18S) and 33 r-proteins (*Ben-Shem et al., 2011*; *Klinge et al., 2011*; *Rabl et al., 2011*).

Assembly of the eukaryotic ribosome takes place in three distinct cellular territories: the nucleolus, the nucleoplasm and the cytoplasm (*Woolford and Baserga, 2013*; *Gerhardy et al., 2014*). RNA polymerase I drives production of the 35S pre-rRNA transcript in the nucleolus, which initiates the assembly process. The emerging 35S pre-rRNA transcript undergoes co-transcriptional modification and processing (*Osheim et al., 2004*; *Kos and Tollervey, 2010*), and associates primarily with 40S subunit r-proteins and ~50 assembly factors to form the earliest pre-ribosome, the 90S (*Dragon et al., 2002*; *Grandi et al., 2002*; *Schäfer et al., 2003*). Cleavage of 35S pre-rRNA releases the pre-40S particle, permitting the remaining pre-rRNA to associate with r-proteins of the 60S subunit and ~200 additional assembly factors to undergo further maturation and pre-rRNA processing (*Fatica et al., 2002*; *Grandi et al., 2002*; *Nissan et al., 2002*). Nuclear maturation of pre-ribosomal particles also requires the release of assembly factors, a process thought to require >50 energy consuming enzymes (*Strunk and Karbstein, 2009*; *Kressler et al., 2010*). Export competent pre-ribosomal particles are separately

**eLife digest** The production of a protein in a cell starts with a region of DNA being transcribed to produce a molecule of messenger RNA. A large molecular machine called ribosome then reads the information in the messenger RNA molecule to produce a protein. Ribosomes themselves are made of RNA and several different proteins called r-proteins. The construction of a ribosome starts with the assembly of a pre-ribosome inside the cell nucleus, and the ribosome is completed in the cytosol of the cell.

A yeast cell will divide about 30 times during its lifetime, and before each division event a single yeast cell needs to import about 14 million r-proteins into its nucleus in order to make about 200,000 ribosomes. However, many details of this process are mysterious. In particular, many r-proteins are known to be unstable: meaning that, left to their own devices, r-proteins are highly likely to aggregate, which would prevent them becoming part of a ribosome.

Now, Schütz et al. have figured out how a carrier protein called Tsr2 makes sure that an r-protein called eS26 does indeed become part of a ribosome. The human disorder known as Diamond-Blackfan anemia is caused by a mutation in the gene for eS26.

The eS26 proteins are ferried to the cell nucleus on specialized transport vehicles. Schütz et al. have now shown that the Tsr2 carrier protein unloads the r-protein from the transport vehicle in the nucleus, and then binds it. This means that the r-protein does not form an aggregate. Finally, the Tsr2 carrier protein transfers the r-protein to the pre-ribosome. This is the first time that a carrier protein that unloads an r-protein cargo from its transport vehicle, to ensure safe delivery to the pre-ribosome, has been identified.

transported through nuclear pore complexes (NPCs) into the cytoplasm by multiple export factors. In yeast, export factors include the exportin Xpo1, which recognizes nuclear export sequences (NESs) in a RanGTP-dependent manner, and additional factors (*Tschochner and Hurt, 2003*). Export factors bind pre-ribosomal particles and interact simultaneously with FG-repeat nucleoporins lining the NPC channel (*Gadal et al., 2001*; *Johnson et al., 2001*; *Oeffinger et al., 2004*; *Bradatsch et al., 2007*; *Yao et al., 2008*, *2010*; *Hackmann et al., 2011*; *Altvater et al., 2012*; *Bassler et al., 2012*; *Faza et al., 2012*; *Occhipinti et al., 2013*).

Following export, pre-ribosomal particles undergo final maturation prior to initiating translation. This involves the release of shuttling assembly factors, transport factors, incorporation of the remaining r-proteins and final pre-rRNA processing (*Panse and Johnson, 2010*; *Panse, 2011*). Within the pre-40S particle, immature 20S pre-rRNA is endonucleolytically cleaved into mature 18S rRNA by the nuclease Nob1 rendering the subunit translation competent (*Fatica et al., 2004*; *Lamanna and Karbstein, 2009*; *Pertschy et al., 2009*). Although, Nob1 is recruited to 40S pre-ribosomes in the nucleus, it is activated in the cytoplasm within an 80S-like pre-ribosomal particle formed upon interaction with a mature 60S subunit (*Lebaron et al., 2012*; *Strunk et al., 2012*). Additionally, multiple conserved ATPases Prp43, Rio2, Rli1 and Fap7, the Prp43-activator Pfa1, the kinase Rio1, the assembly factor Ltv1 and the r-protein uS11 (yeast Rps14) are implicated in this cleavage step (*Geerlings et al., 2003*; *Vanrobays et al., 2003*; *Jakovljevic et al., 2004*; *Granneman et al., 2005*; *Pertschy et al., 2009*; *Strunk et al., 2012*; *Hellmich et al., 2013*). Despite the identification of a plethora of factors and their general order of action, how nuclear and cytoplasmic assembly steps are coordinated remains largely unknown.

In addition to the tremendous energy required to assemble ribosomes, this process also accounts for the major proportion of the nucleocytoplasmic transport in a growing yeast cell (*Rout et al., 1997*; *Sydorskyy et al., 2003*). All mRNAs encoding r-proteins must be exported into the cytoplasm, where translation occurs. Nearly all newly synthesized r-proteins are then imported into the nucleus. In yeast, the importin Kap123 has been shown to be an important mediator of r-protein import, but the related importin Pse1 can functionally substitute Kap123 in vivo (*Rout et al., 1997*; *Schlenstedt et al., 1997*). Unlike other cargos, r-proteins contain large unstructured regions that form intricate interactions with rRNA within the mature ribosome and are prone to non-specific interactions with nucleic acids, aggregation and proteolytic degradation in their non-assembled state (*Jäkel and Görlich, 1998*; *Jäkel et al., 2002*; *Klinge et al., 2011*; *Rabl et al., 2011*). In contrast to typical protein transport events, nuclear import of r-proteins and subsequent transfer to the ribosome production site pose logistical challenges.

In addition to their transport role, importins have been implicated to chaperone basic r-proteins during their transport to the nucleus (*Jäkel et al., 2002*). How these intrinsically unstable and aggregation-prone proteins are targeted to assembling pre-ribosomal particles after dissociating from importins remains unclear.

Here, we report the discovery of a carrier Tsr2 that coordinates transfer of the eukaryote specific r-protein eS26 (yeast Rps26; *Ban et al., 2014*) after nuclear import to the assembling 90S pre-ribosome. Tsr2 extracts eS26 from its importins to terminate its import process. Hereby, we reveal an atypical RanGTP-independent mechanism to dissociate an importin:cargo complex. Tsr2 binds and protects the released eS26 from aggregation and proteolysis thereby ensuring its safe transfer to the 90S pre-ribosome. Our data raise the possibility of a yet unidentified fleet of carriers that securely link the nuclear import machinery with the ribosome assembly pathway.

## Results

### Tsr2 is required for cytoplasmic processing of 20S pre-rRNA to mature 18S rRNA

Previous genome-wide studies revealed a strong accumulation of immature 20S pre-rRNA in a *TSR2* (20 S rRNA accumulation 2) deficient yeast strain (*tsr2Δ*) (*Peng et al., 2003*). Tsr2 is a conserved 23.7 kDa protein (*Figure 1—figure supplement 1A*) with no identified structural homologues that could provide clues into its role in 20S pre-rRNA processing. To dissect the function of Tsr2, we generated a conditional mutant in which the endogenous *TSR2* was placed under the control of the *GAL1* promoter ($P_{GAL1}$-*TSR2*). On repressive glucose media, Tsr2 protein levels were undetectable and the $P_{GAL1}$-*TSR2* strain was severely impaired in growth compared to a wild-type (WT) strain between 20–37°C (*Figure 1A*).

Next, we localized Tsr2 using an integrated C-terminal -GFP and -TAP tag at the genomic locus. These cell-biological studies revealed that both fusion proteins predominantly localize to the nucleus (*Figure 1B*). A similar location for the Tsr2-3xGFP fusion protein (expressed from a CEN plasmid under its natural promoter and terminator regions) was observed in a Tsr2-depleted strain. The strains expressing the various fusion proteins were not impaired in growth (*Figure 1A*) suggesting that addition of the -GFP, -TAP, and -3xGFP tags did not affect Tsr2 function. We conclude that Tsr2 mainly localizes to the nucleus.

The location of Tsr2 led us to test whether the accumulation of 20S pre-rRNA in *tsr2Δ* cells *Peng et al. (2003)* is due to impaired nuclear export of pre-40S subunits. To this end, we monitored localization of 40S subunits in Tsr2-depleted cells using the established reporter uS5-GFP (yeast Rps2-GFP; *Milkereit et al., 2001*). We used the *yrb2Δ* mutant, which is specifically impaired in pre-40S subunit export, as a control (*Moy and Silver, 2002*). As expected, the *yrb2Δ* mutant showed a nuclear accumulation of uS5-GFP, in contrast to WT, which displayed cytoplasmic localization of this reporter (*Figure 1—figure supplement 1B*). Surprisingly, $P_{GAL1}$-*TSR2* cells grown on glucose also showed cytoplasmic uS5-GFP localization (*Figure 1—figure supplement 1B*), indicating no apparent impairment in nuclear export of pre-40S subunits.

The data above raised the possibility that cytoplasmic processing of 20S pre-rRNA is impaired in Tsr2-depleted cells. To this end, we monitored the localization of the 5′ portion of the internal transcribed spacer 1 (ITS1) that is present within immature 20S pre-rRNA, but not in mature 18S rRNA, by fluorescence in situ hybridization (FISH). In a WT strain, due to efficient nuclear export of pre-40S subunits, Cy3-ITS1 (red) is detectable only in the nucleolus (*Figure 1C*). After nuclear export, ITS1 is cleaved from 20S pre-rRNA by the endonuclease Nob1 and degraded by the exonuclease Xrn1 (*Stevens et al., 1991*; *Moy and Silver, 2002*). Tsr2-depleted cells exhibited strong cytoplasmic accumulation of Cy3-ITS1 (*Figure 1C*), indicating that cytoplasmic processing is impaired in these cells.

Two studies proposed that 20S pre-rRNA processing occurs within an 80S-like particle formed via interaction between a mature 60S subunit and a pre-40S subunit in the cytoplasm (*Lebaron et al., 2012*; *Strunk et al., 2012*). One possibility is that formation of this particle is impaired in Tsr2-depleted cells, thereby indirectly interfering with 20S pre-rRNA processing. To test this, we performed polysome analyses. Cell extracts from WT and Tsr2-depleted cells prepared under polysome preserving conditions were analyzed by sucrose gradient centrifugation. In agreement with a role in the 40S biogenesis pathway, the polysome profile of Tsr2-depleted cell extracts revealed strongly reduced levels of free 40S subunits and polysomes (*Figure 1D*, top panel). Northern analyses revealed that mature 25S rRNA

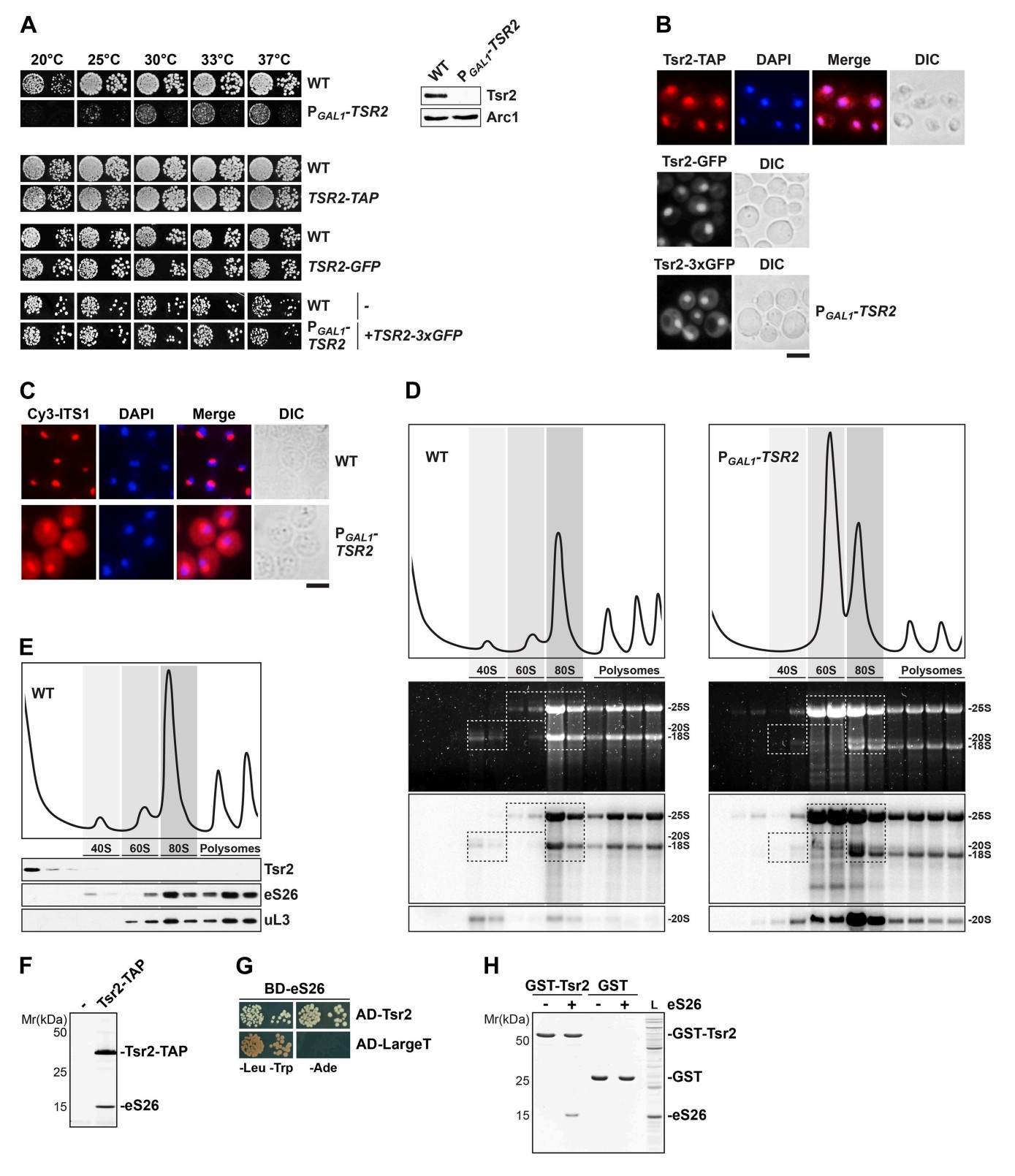

**Figure 1**. Tsr2 is required for cytoplasmic processing of 20S pre-rRNA to mature 18S rRNA, and directly binds eS26. (**A**) Tsr2-TAP, Tsr2-GFP and Tsr2-3xGFP cells are not impaired in growth. Left panel: indicated strains were spotted on glucose containing media in 10-fold dilutions and grown at indicated temperatures for 3–7 days. Right panel: Tsr2 protein levels in whole cell extracts derived from the indicated strains were determined by
*Figure 1. Continued on next page*

Figure 1. Continued

Western analyses using α-Tsr2 antibodies. Protein levels of Arc1 served as loading control. (**B**) Tsr2 localizes predominantly to the nucleus. The Tsr2-TAP and the Tsr2-GFP strain and the $P_{GAL1}$-$TSR2$ strain containing a centrometric plasmids encoding Tsr2-3xGFP were grown at 30°C to mid-log phase. Localization of Tsr2-TAP was visualized by indirect immunofluorescence microscopy using polyclonal α-TAP antibody (red). Nuclear and mitochondrial DNA was stained with DAPI (blue). Localization of Tsr2-GFP and Tsr2-3xGFP was analyzed by fluorescence microscopy. Scale bar = 5 µm. (**C**) Tsr2-deficient cells accumulate immature 20S pre-rRNA in the cytoplasm. WT and $P_{GAL1}$-$TSR2$ cells were grown at 30°C in glucose containing media to mid-log phase. Localization of 20S pre-rRNA was analyzed by FISH using a Cy3-labeled oligonucleotide complementary to the 5' portion of ITS1 (red). Nuclear and mitochondrial DNA was stained with DAPI (blue). Scale bar = 5 µm. (**D**) Tsr2-depleted cells accumulate 80S-like particles. WT and $P_{GAL1}$-$TSR2$ cells were grown at 30°C in glucose containing media to mid-log phase. Cell extracts were prepared after cycloheximde treatment to preserve polysomes and subjected to sedimentation centrifugation on 7–50% sucrose gradients. Polysome profiles at OD$_{254nm}$ were recorded and the peaks for 40S and 60S subunits, 80S ribosomes and polysomes are indicated (top panels). The gradients were fractionated and the RNA was extracted, separated on a 2% Agarose gel, stained with GelRed (Biotium, middle panels) and subsequently analyzed by Northern Blotting using probes against indicated rRNAs (bottom panels). Exposure times for phosphoimager screens were 20 min for 25S and 18S rRNA, and 3–4 hr for 20S pre-rRNAs. (**E**) Tsr2 does not co-sediment with 40S subunits. WT cells were grown at 30°C to mid-log phase, extracts were prepared and fractionated as described in (**D**). The polysome profile at OD$_{254nm}$ is shown in the upper panel. The peaks for 40S and 60S subunits, 80S ribosomes and polysomes are indicated. The gradient was fractionated, TCA precipitated and the protein content was assessed by Western analyses using the indicated antibodies. (**F**) Tsr2-TAP co-enriches the r-protein eS26. Tsr2-TAP was isolated by tandem affinity purification and the Calmodulin-eluate was separated by 4–12% gradient SDS-PAGE and analyzed by Silver staining. The indicated proteins were identified by mass spectrometry. (**G**) Tsr2 interacts with eS26 in a yeast two-hybrid assay. Plasmids encoding the indicated *GAL4* DNA-binding domain *(BD)* and *GAL4* activation domain *(AD)* fusion proteins were transformed into the yeast reporter strain NMY32. Transformants were spotted in 10-fold serial dilutions onto SDC-Leu-Trp (-Leu-Trp) or SDC-Ade (−Ade) and incubated at 30°C for 4 days. Growth on SDC-Ade indicates a strong two-hybrid interaction. The SV40 Large T antigen served as negative control for these analyses. (**H**) Tsr2 directly binds eS26 in vitro. GST-Tsr2 was immobilized on Glutathione Sepharose before incubation with an *E. coli* lysate containing recombinant eS26. After incubation, bound proteins were eluted by SDS sample buffer, separated by SDS-PAGE and visualized by Coomassie Blue staining. L = input.

The following figure supplement is available for figure 1:

**Figure supplement 1**. Tsr2 and eS26 depletion does not impair pre-40S nuclear export.

and immature 20S pre-rRNA co-peak (*Figure 1D*, bottom panel), indicating accumulation of 80S-like particles, similar to the one seen upon Fap7-depletion (*Granneman et al., 2005*; *Strunk et al., 2012*). Thus, pre-40S subunits that are exported into the cytoplasm in Tsr2-depleted cells interact with mature 60S subunits, but fail to undergo 20S pre-rRNA processing. We conclude that Tsr2 is required for cytoplasmic maturation of pre-40S subunits.

## Tsr2 directly binds the eukaryote specific r-protein eS26

Next, we analyzed the sedimentation behavior of Tsr2 on sucrose density gradients. Cell extracts from WT cells were subjected to polysome analyses. The gradient was fractionated and analyzed by Western analyses. Unexpectedly, Tsr2 did not co-sediment with the 40S peak or with heavier fractions, but was found exclusively in lighter fractions at the top of the gradient (*Figure 1E*). These data indicate that Tsr2 does not stably associate with pre-ribosomal particles in the 40S biogenesis pathway.

To identify interaction partners of Tsr2, we isolated Tsr2-TAP. In agreement with the sedimentation studies above, Tsr2-TAP did not isolate a pre-40S particle. Instead, Tsr2-TAP co-enriched stoichiometric amounts of the eukaryotic specific r-protein eS26 (*Figure 1F*; *Peng et al., 2003*). Further, yeast two-hybrid analysis revealed a strong interaction between Tsr2 and eS26, as determined by growth on stringent adenine deficient media (*Figure 1G*). In vitro binding studies using recombinant proteins showed that eS26 and Tsr2 formed a robust complex (*Figure 1H*). We conclude that eS26 directly binds Tsr2.

## eS26 is required for cytoplasmic processing of 20S pre-rRNA

In budding yeast, two non-essential genes, *RPS26A* and *RPS26B*, encode the r-protein eS26. To investigate the phenotypes of *RPS26* deficiency, we created a conditional double mutant in which the endogenous promoter of *RPS26A* in the *rps26bΔ* strain was replaced with the *GAL1* promoter ($P_{GAL1}$-*RPS26A*). Consistent with an essential function of eS26 in yeast, the $P_{GAL1}$-*RPS26Arps26bΔ* strain did not grow on repressive glucose containing medium (*Figure 2A*). Using this strain, we investigated whether eS26 is required for nuclear export of pre-40S subunits and/or cytoplasmic 20S pre-rRNA processing by monitoring the localization of uS5-GFP and Cy3-ITS1. eS26-depletion did not induce nuclear accumulation of uS5-GFP (*Figure 1—figure supplement 1B*), indicating no apparent impairment in pre-40S subunit nuclear export. However, these cells showed a strong cytoplasmic accumulation of Cy3-ITS1 (*Figure 2B*),

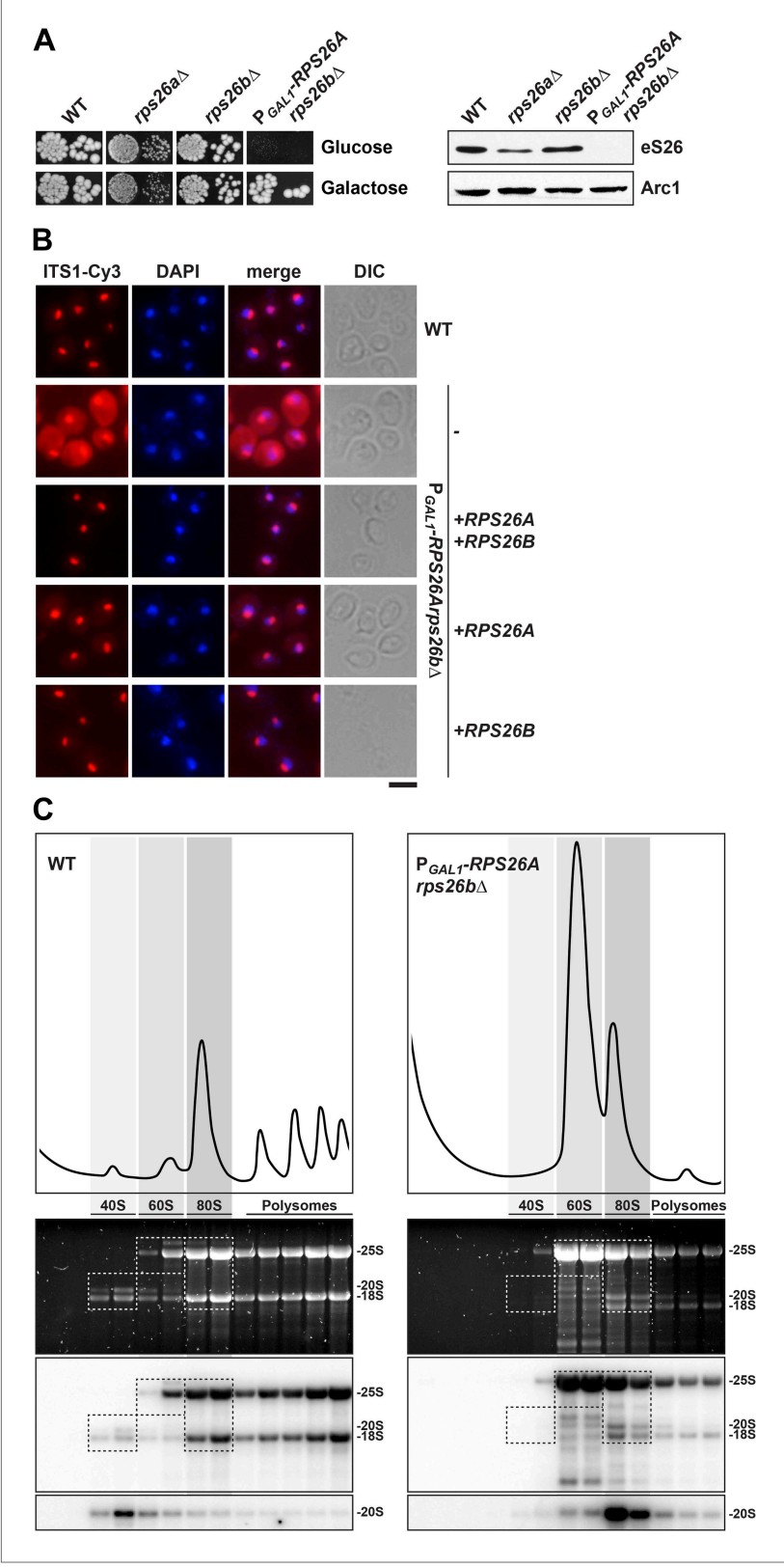

**Figure 2**. eS26 is required for cytoplasmic processing of immature 20S pre-rRNA to mature 18S rRNA. (**A**) eS26 is essential for viability in yeast. Left panel: WT, *rps26aΔ, rps26bΔ* and the conditional mutant P*GAL1*-*RPS26Arps26bΔ* were spotted in 10-fold dilutions on galactose and repressive glucose containing media and grown at 30°C for 2–4

*Figure 2. Continued on next page*

*Figure 2. Continued*

days. Right panel: protein levels of eS26 in whole cell extracts of indicated strains were determined by Western analyses using α-eS26 antibodies. Arc1 protein levels served as loading control. (**B**) eS26-depleted cells accumulate immature 20S pre-rRNA in the cytoplasm. P$_{GAL1}$-*RPS26Arps26bΔ* cells transformed with indicated plasmids were grown in glucose containing liquid media at 37°C to mid-log phase. Localization of 20S pre-rRNA was analyzed by FISH using a Cy3-labeled oligonucleotide complementary to the 5′ portion of ITS1 (red). Nuclear and mitochondrial DNA was stained with DAPI (blue). Scale bar = 5 μm. (**C**) eS26-depleted cells accumulate 80S-like particles. The indicated strains were grown in glucose containing liquid media at 30°C to mid-log phase. Cell extracts were prepared after cycloheximide treatment and subjected to sedimentation centrifugation on 7–50% sucrose density gradients. Polysome profiles were recorded at OD$_{254nm}$ (top panels). The peaks for 40S and 60S subunits, 80S ribosomes and polysomes are indicated. Sucrose gradients were fractionated, the RNA was extracted, separated on a 2% Agarose gel, stained with GelRed (Biotium, middle panels) and subsequently analyzed by Northern blotting using probes against the indicated rRNAs (bottom panels). Exposure times for phosphoimager screens were 20 min for 25S and 18S rRNA, and 3–4 hr for 20S pre-rRNAs.

indicating impairment in final 20S pre-rRNA processing. Further, polysome analyses of eS26-depleted cell extracts revealed strongly reduced levels of free 40S subunits (*Figure 2C*, top panel). Northern analyses revealed that mature 25S rRNA and immature 20S pre-rRNA co-peaked (*Figure 2C*, bottom panel), indicating an accumulation of 80S-like particles. Thus, as observed in Tsr2-depleted cells, eS26-depleted cells contain pre-40S subunits that fail to process 20S pre-rRNA in the cytoplasm. Based on these data we conclude that eS26 is required for cytoplasmic maturation of pre-40S subunits.

## eS26 is recruited to the earliest pre-ribosomal particle, the 90S

The robust interaction between the predominantly nuclear localized Tsr2 and eS26 prompted us to investigate at which stage eS26 is recruited to pre-40S subunits. To address this, we isolated pre-ribosomal particles at different maturation stages along the 40S biogenesis pathway (*Grandi et al., 2002*; *Schäfer et al., 2003*). Noc4-TAP purifies the earliest precursor of the pre-40S subunit, the 90S pre-ribosome; Enp1-TAP purifies both the 90S and early pre-40S subunits; Rio2-TAP purifies a late pre-40S subunit containing immature 20S pre-rRNA; and Asc1-TAP purifies a 40S subunit containing mature 18S rRNA and devoid of late assembly factors (*Figure 3A*, *Figure 3—figure supplement 1A*). Co-enrichment of eS26 with pre-ribosomal particles was assessed by (1) Western analyses using antibodies that recognize eS26 and (2) selected reaction monitoring mass spectrometry (SRM-MS). SRM-MS is a reliable tool that overcomes stochastic under sampling of peptides, a critical deficit in shotgun mass spectrometry which complicates the reproducible detection and precise quantitation of proteins in a complex mixture (*Picotti and Aebersold, 2012*). SRM relies on the development of specific mass spectrometric-based assays for every target protein and their subsequent application to the relative or absolute quantification within multiple biological samples. We developed a set of SRM assays that enabled us to simultaneously monitor the co-enrichment of eS26 and different r-proteins: uS7 (Rps5), eS28 (Rps28), eS1 (Rps1) and uS11 (Rps14) (*Figure 3B*) with multiple pre-ribosomal particles. Both Western and SRM analyses revealed that eS26 co-enriches efficiently with the earliest ribosomal precursor, the 90S, and different pre-ribosomes along the 40S maturation pathway (*Figure 3A,B*). The Western signal for eS26 on the 90S pre-ribosome (Noc4-TAP) is specific since no association was detected with the earliest 60S pre-ribosome (Ssf1-TAP) (*Figure 3—figure supplement 1B*).

To support these biochemical data, we performed a complementary cell-biological experiment. If eS26 were targeted to the 90S pre-ribosome, then impairment in pre-40S subunit export should result in its accumulation in the nucleus. To monitor eS26 localization in vivo, we tagged *RPS26A* with GFP at the C-terminus (eS26-GFP) in WT and *yrb2Δ* cells at the genomic locus. Unlike the *rps26aΔ* mutant, the *RPS26A-GFP* strain was not impaired in growth at 20°C and 37°C indicating that addition of GFP does not impair its function on the 40S subunit (*Figure 3C*, left panel). As expected, WT cells displayed a strong cytoplasmic localization of eS26-GFP (*Figure 3C*, right panel). In contrast, in *yrb2Δ* cells eS26-GFP accumulated in the nucleus (*Figure 3C*, right panel). Together all these data suggest that eS26 is transported to the nucleus for loading on the 90S pre-ribosome.

Consistent with the sedimentation studies and direct binding to only eS26 (*Figure 1*), Tsr2 did not detectably co-enrich with affinity purified pre-ribosomal particles in the 40S maturation pathway (*Figure 3A*). This lack of co-enrichment was not due to altered protein levels in the different TAP strains, since Western analyses of whole cell extracts revealed that Tsr2 was expressed at WT levels

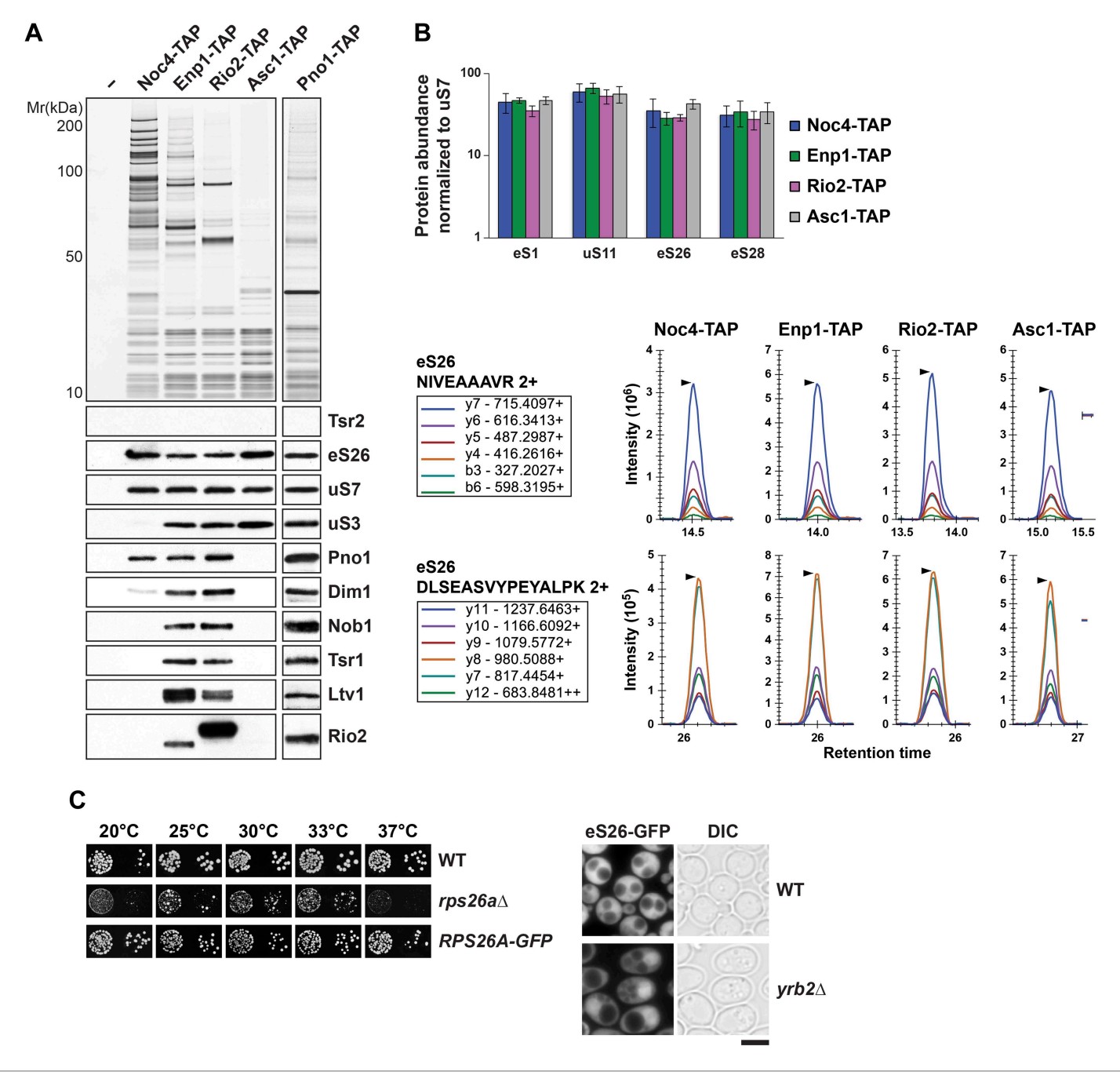

**Figure 3**. eS26 is incorporated into the earliest pre-ribosome, the 90S. (**A**) eS26 co-enriches with pre-ribosomal particles along the 40S maturation pathway. Pre-ribosomal particles in the 40S maturation pathway were purified using the indicated TAP-tagged baits. Calmodulin-eluates were analyzed by Silver staining and Western analyses using the indicated antibodies. The r-protein uS7 served as loading controls for the TAPs. (**B**) SRM-MS reveals co-enrichment of eS26 with pre-ribosomal particles. Upper panel: the relative abundance of different r-proteins was normalized to uS7 levels in the indicated TAP purifications (three independent biological replicates). The error bars show the standard deviation. Lower panel: the intensity of different transitions (listed in the box) of two specific peptides of eS26 was determined by SRM mass spectrometry in the indicated TAP purifications. (**C**) eS26-GFP accumulates in the nucleus in a *yrb2Δ* strain. Left panel: WT, *rps26aΔ* and *RPS26A-GFP* cells were spotted in 10-fold dilutions and grown at indicated temperatures for 3–7 days. Right panel: WT and *yrb2Δ* cells expressing eS26-GFP were grown in glucose containing liquid media to mid-log phase at 20°C. Localization of eS26-GFP was monitored by fluorescence microscopy. Scale bar = 5 µm.

The following figure supplement is available for figure 3:

**Figure supplement 1**. Tsr2 and eS26 protein levels in the indicated TAP strains and levels of 20S pre-rRNA and 18S rRNA in the indicated TAP purified particles.

(*Figure 3—figure supplement 1C*). Altogether, these results suggest that there are at least two populations of eS26 in vivo, one bound to ribosomes and another bound to Tsr2.

## eS26 is imported primarily by Kap123 and Kap104 into the nucleus

We next investigated how eS26 is imported into the nucleus prior to its incorporation into the 90S pre-ribosome. In yeast, the most abundant importin Kap123 transports various r-proteins into the nucleus (*Rout et al., 1997*; *Schlenstedt et al., 1997*). However, r-proteins also utilize additional importins, including Pse1, Kap104, Sxm1 and Nmd5 (*Rout et al., 1997*; *Sydorskyy et al., 2003*). We investigated the interaction between eS26 and all yeast importins in vitro. These studies revealed that the importins Kap123, Kap104 and Pse1 efficiently bound eS26 (*Figure 4A*). A very weak interaction was observed between Sxm1, Kap95 and Nmd5 and eS26, and no binding was observed with the remaining importins (*Figure 4—figure supplement 1*). In contrast, none of the importins bound to either Tsr2 or the Tsr2:eS26 complex (*Figure 4A*, *Figure 4—figure supplement 1*), indicating that eS26 alone specifically interacts with importins.

To verify our interaction data in vivo, we monitored nuclear uptake of eS26 in WT cells and in different importin mutants. The r-protein eS26 is assembled into the 90S pre-ribosome and is then rapidly transported to the cytoplasm as part of the 40S pre-ribosome. To investigate eS26 nuclear uptake in vivo we uncoupled its import from its export. Structural analyses of the 40S subunit showed that the N-terminus of eS26 is embedded within the rRNA framework (*Figure 4—figure supplement 2A*; *Rabl et al., 2011*). We fused GFP to the N-terminus of eS26 with the aim to impair its incorporation into the 90S pre-ribosome. Sucrose gradient analyses showed that GFP-eS26 co-sediments only in lighter fractions at the top of the gradient suggesting that it is not incorporated into pre-ribosomes (*Figure 4—figure supplement 2B*). In vitro binding studies showed that like eS26, GFP-eS26 interacts with Kap123, Kap104 and Pse1 (*Figure 4—figure supplement 2C*). Thus, the GFP-eS26 fusion protein is functional to recruit the import machinery, although it does not complement the eS26-depleted strain (*Figure 4—figure supplement 2D*). Further, GFP-eS26 directly binds Tsr2 (*Figure 4—figure supplement 2C*) and importantly, like eS26, GFP-eS26 is degraded upon Tsr2-depletion (*Figure 4—figure supplement 2E*). We exploited the GFP-eS26 fusion protein as a tool to monitor the nuclear uptake of eS26 in different importin mutants. Consistent with in vitro binding assays, nuclear uptake of GFP-eS26 was reduced in *kap123Δ* and *kap104Δ* cells (*Figure 4B*), indicating that eS26 import requires these importins. Nuclear localization of GFP-eS26 in the *pse1-1 ts* mutant at restrictive temperature remained unaffected (*Figure 4B*) indicating that impairment of this importin alone does not inhibit the nuclear import of eS26. The *pse1-1 kap123Δ* mutant showed only a slight increase in cytoplasmic staining of GFP-eS26 (*Figure 4B*). Nuclear import of GFP-eS26 was unaffected in the *kap114Δ sxm1Δ* double mutant and *sxm1Δ kap120Δ nmd5Δ* triple mutant (*Figure 4B*).

Next, we investigated which region of eS26 contributes to its nuclear uptake. For this, we monitored the localization of different truncated versions of eS26 fused to -GFP at the N-terminus. These cell-biological analyses revealed that the Zn²⁺-binding domain is required for efficient nuclear uptake of eS26 (*Figure 4—figure supplement 2F*). If eS26 were imported into the nucleus in complex with Tsr2, then we reasoned that depletion of eS26 would induce Tsr2 mislocalization to the cytoplasm. However, localization of Tsr2-3xGFP was not affected upon eS26-depletion (*Figure 4C*). These studies together with the observation that Tsr2:eS26 complex is unable to recruit importins argue against the idea that eS26 is transported to the nucleus in complex with Tsr2. We conclude that Kap123 and Kap104 target eS26 to the nucleus and that Tsr2 is not a component of this import complex.

## Kap123 targets Tsr2 to the nucleus

Next, we investigated how Tsr2 is targeted to the nucleus in vivo. For this, we monitored the location of Tsr2-3xGFP in different importin mutants. We found that Tsr2-3xGFP mislocalizes to the cytoplasm in the *kap123Δ* mutant, but not in other importin mutants for e.g. *kap104Δ* and *pse1-1* (*Figure 4C*). Thus, Kap123 seems to be the major import receptor for Tsr2. However, we did not observe a direct interaction between Tsr2 and Kap123 or any other importin in vitro (*Figure 4A*, *Figure 4—figure supplement 1*). One possibility could be that import of Tsr2 by Kap123 is regulated by post-translational modification. Alternatively, Tsr2 might be transported into the nucleus via a 'piggy bag' mechanism bound to another yet unknown Kap123 cargo. We can exclude the possibility that eS26 serves as an adaptor to import Tsr2 since (1) Tsr2 does not mislocalize to the cytoplasm in a eS26-depleted strain (*Figure 4C*) and (2) in vitro binding assays show that the Tsr2:eS26 complex does not interact with Kap123 (*Figure 4A*).

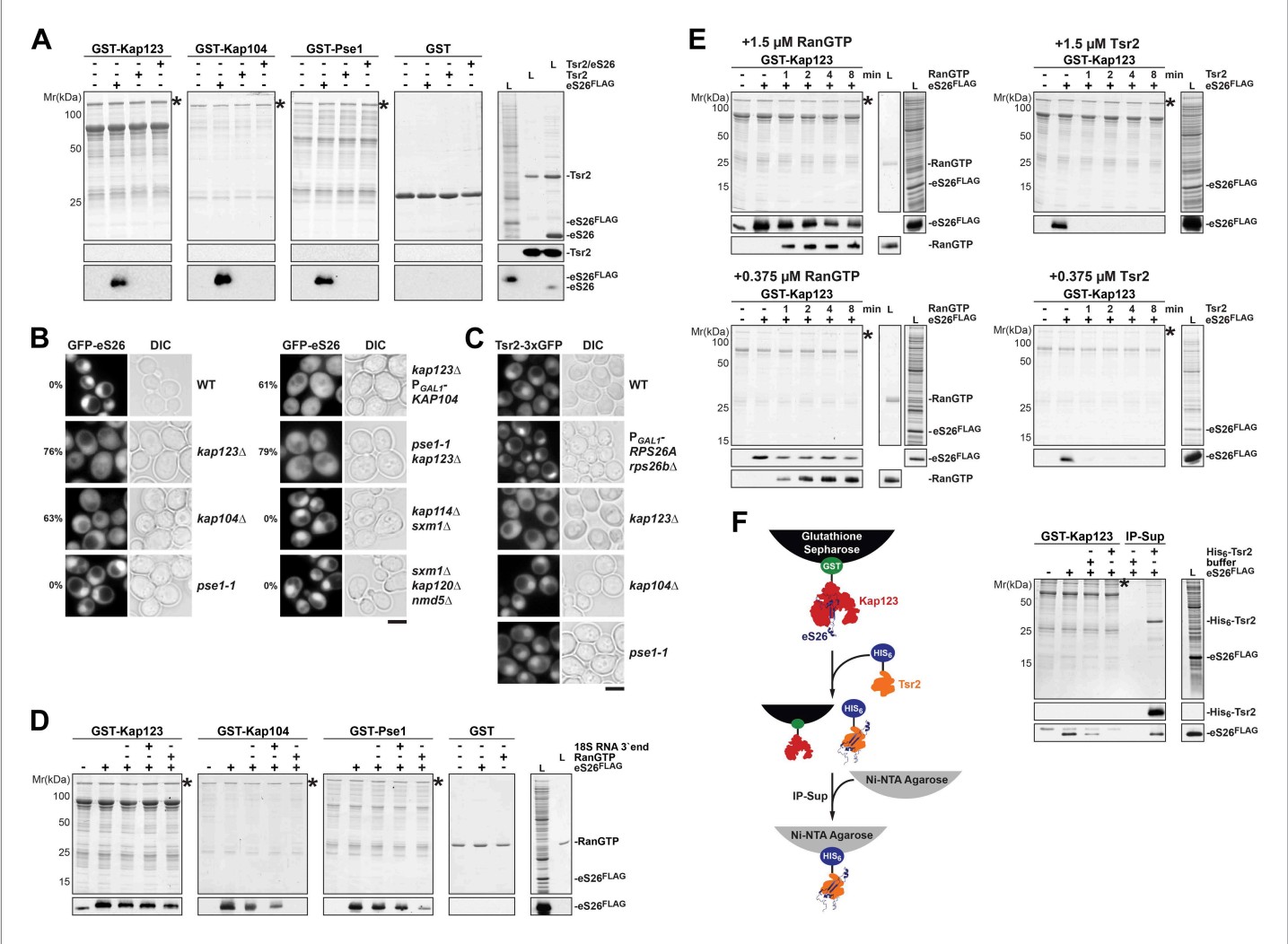

**Figure 4**. Kap123, Kap104 and Pse1 transport eS26 to the nucleus. (**A**) eS26, but not Tsr2:eS26 or Tsr2, interacts with Kap123, Kap104 and Pse1. Recombinant, GST-Kap123, GST-Kap104, GST-Pse1 and GST alone were immobilized on Glutathione Sepharose and incubated with purified 3.4 µM Tsr2, 4 µM Tsr2:eS26, or *E. coli* lysate containing ~4 µM eS26[FLAG] in PBSKMT combined with competing *E. coli* lysates for 1 hr at 4°C. After washing with PBSKMT, bound proteins were eluted in SDS sample buffer and separated by SDS-PAGE. Proteins were visualized by Coomassie Blue staining or Western analyses using indicated antibodies. L = input. GST-tagged importins are indicated with asterisks. (**B**) Nuclear uptake of GFP-eS26 is impaired in *kap123Δ* and *kap104Δ* mutants. Strains expressing GFP-eS26 were grown in synthetic media at 25°C (*ts*-mutants: *pse1-1* and *kap104Δ*) or 30°C to mid-log phase. *Ts*-mutant strains were then shifted to 37°C for 4 hr and localization of GFP-eS26 was analyzed by fluorescence microscopy. Percentage of cells displaying cytoplasmic mislocalization of the GFP-eS26 fusion is indicated. Scale bar = 5 µm. (**C**) Tsr2-3xGFP is targeted to the nucleus by Kap123. Importin mutant strains expressing Tsr2-3xGFP were grown in synthetic media at 25°C (*ts*-mutants: *pse1-1* and *kap104Δ*) or 30°C to mid-log phase. *Pse1-1* and *kap104Δ* cells were then shifted to 37°C for 4 hr. P[GAL1]-*RPS26Arps26bΔ* cells containing Tsr2-3xGFP were grown for 15 hr in glucose contain-ing media. Localization of Tsr2-3xGFP was analyzed by fluorescence microscopy. Scale bar = 5 µm. (**D**) RanGTP (His[6]-Gsp1Q71L-GTP) does not efficiently release eS26 from Kap123 and Pse1. GST-importin:eS26[FLAG] complexes immobilized on Glutathione Sepharose were incubated with either buffer alone or with 1.5 µM RanGTP or 3 nM 3'-end of 18S rRNA for 1 hr at 4°C. Washing, elution, and visualization were performed as in (**A**). GST-tagged importins are indicated with asterisks. (**E**) Tsr2 efficiently dissociates the Kap123:eS26[FLAG] complex. The GST-Kap123: eS26[FLAG] complex immobilized on Glutathione Sepharose was incubated with either buffer alone or with 1.5 µM or 375 nM RanGTP or 1.5 µM or 375 nM Tsr2. Samples were withdrawn at the indicated time points. Washing, elution, and visualization were performed as in (**A**). GST-tagged Kap123 is indicated with an asterisk. (**F**) eS26 stably associates with Tsr2 after its release from Kap123. Left panel indicates the experimental setup as flowchart. Immobilized GST-Kap123:eS26[FLAG] complex was incubated with 1.5 µM His[6]-Tsr2 or buffer alone. As shown in the flowchart, the supernatant was incubated with Ni-NTA Agarose for 1 hr at 4°C (IP-Sup). Washing, elution, and visualization were performed as in (**A**). GST-tagged Kap123 is indicated with an asterisk.

The following figure supplements are available for figure 4:

**Figure supplement 1**. eS26, but not Tsr2:eS26 or Tsr2, interacts with importins.

*Figure 4. Continued on next page*

*Figure 4. Continued*

**Figure supplement 2**. GFP-eS26 binds to importins and Tsr2 but is not incorporated into pre-ribosomes.

**Figure supplement 3**. Tsr2 efficiently releases the conserved eS26 from importins.

**Figure supplement 4**. RanGTP and Tsr2 do not release eS31, eS8 and uS14 from Kap123.

## Tsr2 dissociates importin:eS26 complexes in a RanGTP-independent manner

After transport of an importin:cargo complex into the nucleus, RanGTP binds to the N-terminal region of the importin, triggering cargo release and allowing recycling of the importin to participate in subsequent import cycles (*Lee et al., 2005*; *Cook et al., 2007*; *Kobayashi and Matsuura, 2013*). To test whether the release of eS26 from the importins is RanGTP-dependent, we performed in vitro dissociation assays. A pre-formed importin:eS26 complex was incubated with 1.5 μM Gsp1Q71L-GTP (equivalent to the human RanQ69L mutant that cannot efficiently hydrolyze GTP, hereafter Gsp1Q71L-GTP is termed RanGTP; *Bischoff et al., 1994*; *Maurer et al., 2001*). Although RanGTP was able to dissociate the Kap104:eS26 complex, and partially dissociate the Pse1:eS26 complex, we did not observe dissociation of the Kap123:eS26 complex even after 1 hr incubation (*Figure 4D*). It was reported that both RNA and RanGTP are required to release of the mRNA binding proteins Nab2 and Nab4 from Kap104 and the mRNA export factor Npl3 from Mtr10 (*Senger et al., 1998*; *Lee and Aitchison, 1999*). Because eS26 directly interacts with the 3'-end of the 18S rRNA (*Figure 4—figure supplement 3A*, right panel), we tested if this region of the 18S rRNA is required to release eS26 from Kap123. However, eS26 remained stably bound to Kap123 in the presence of this RNA, either alone or in combination with RanGTP (*Figure 4D*, *Figure 4—figure supplement 3B*).

Since the Tsr2:eS26 complex was unable to interact with importins, we tested whether Tsr2 stimulates the release of eS26 from importins. Surprisingly, Tsr2 alone efficiently removed eS26 from Kap123, Pse1 and Kap104 (*Figure 4—figure supplement 3B,C* and data not shown). This release was specific, since only RanGTP, but not Tsr2, was able to remove the 40S assembly factor Slx9 (*Faza et al., 2012*) from the Pse1:Slx9 complex under the same conditions (*Figure 4—figure supplement 3D*). Moreover, Tsr2 specifically releases eS26 from the importin:eS26 complex, since it did not dissociate other tested importin:r-protein complexes (Kap123:uS14, Kap123:eS31 and Kap123:eS8) (*Figure 4—figure supplement 4*).

Since eS26 was inefficiently removed from the Pse1:eS26 complex after 1 hr incubation with RanGTP (*Figure 4D*), we investigated the dissociation kinetics of importin:eS26 complexes in the presence of RanGTP or Tsr2. For this, the importin:eS26 complex was incubated with 1.5 μM of either RanGTP or Tsr2 and the release of eS26 from the importin was monitored over time. We found that the amount of eS26 bound to Kap123, Pse1 or Kap104 was only slightly reduced after 8 min, even though RanGTP was efficiently recruited to the different importins:eS26 complexes (*Figure 4E*, left panel and *Figure 4—figure supplement 3E*, left panel). In contrast, Tsr2 completely removed eS26 from these importins within 1 min incubation (*Figure 4E*, right panel and *Figure 4—figure supplement 3E*, right panel). Notably, even at lower concentrations (375 nM) Tsr2 was able to release eS26 from the importin:eS26 complex (*Figure 4E*, *Figure 4—figure supplement 3E*). Moreover, Tsr2 stably associated with the released eS26 (*Figure 4F*, *Figure 4—figure supplement 3C*).

## Tsr2 shields eS26 from proteolysis and aggregation, and promotes a safe transfer to the 90S pre-ribosome

The observation that Tsr2 is able to extract eS26 from importins, prompted us to investigate whether Tsr2 plays a role in the transfer of eS26 to the assembling pre-ribosome. To test this, we isolated Enp1-TAP, which purifies both the 90S pre-ribosome and an early pre-40S subunit, from WT and Tsr2-depleted cells and assessed co-enrichment of eS26 by Western analyses. Consistent with a role for Tsr2 in supplying eS26 to the 90S pre-ribosome, we found that eS26 does not efficiently co-enrich with Enp1-TAP in Tsr2-depleted cells (*Figure 5A*). This was specific for eS26, since the recruitment of uS7 and uS3 to Enp1-TAP particles was not affected in these cells (*Figure 5A*). This lack of enrichment was

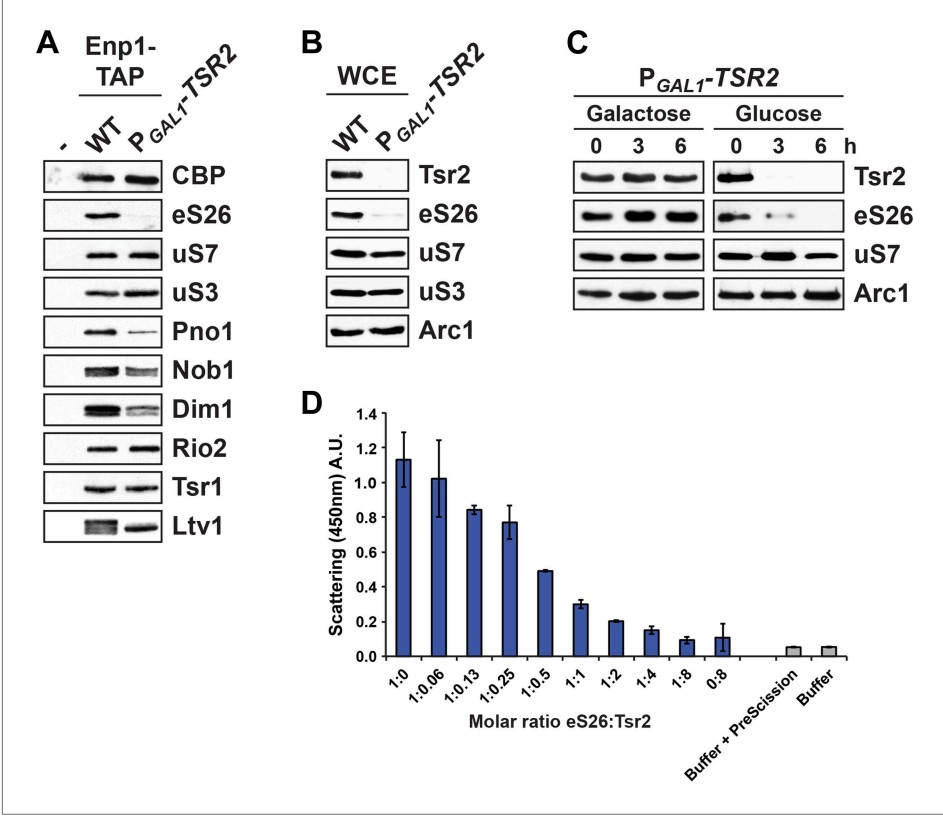

**Figure 5**. Tsr2 shields eS26 from proteolysis and aggregation, and promotes safe transfer to the 90S pre-ribosome. (**A**) Efficient recruitment of eS26 to Enp1-TAP requires Tsr2. Enp1-TAP was isolated from WT and Tsr2-depleted cells. After tandem affinity purification, eluates were separated by 4–12% gradient SDS-PAGE and subjected to Western analyses using indicated antibodies. CBP (α-TAP) levels served as loading control. (**B**) eS26 levels are strongly reduced in Tsr2-depleted cells. Whole cell extracts (WCE) prepared from WT and Tsr2-depleted cells were assessed by Western analyses using antibodies against the indicated proteins. Arc1 protein levels served as loading control. (**C**) Tsr2 protects eS26 from proteolysis in vivo. The conditional mutant strain $P_{GAL1}$-*TSR2* growing on galactose medium was transferred to repressive glucose containing liquid media at 30°C. Cells were withdrawn at the indicated time points and whole cell extracts were prepared. Western analyses were performed to determine the levels of the indicated proteins. Arc1 served as loading control. (**D**) Tsr2 prevents aggregation of recombinant eS26 in vitro. The aggregation assay was performed in a 384-well plate. In each well 33 μM GST-eS26 and a given concentration of Tsr2 (0 up to 266 μM) in PBSKMT was pre-incubated for 1 hr at 4°C (final volume: 90 μl). 250 nM of PreScission protease was added to initiate aggregation. After 1 hr of incubation, the scattering signal of the aggregated eS26 was monitored by a 384-well plate reader by measuring the intensity at 450 nm (Y-axes). Concentration of Tsr2 used in the assay (X-axes) are expressed as a molar ratio of eS26:Tsr2. Four replicates for each well were measured. The error bars show the standard deviation.

due to decreased eS26 protein levels, since Western analyses of whole cell extracts derived from Tsr2-depleted cells revealed strongly reduced eS26 protein levels (***Figure 5B***). These data led us to test whether eS26 becomes susceptible to proteolysis in Tsr2-depleted cells. To this end, we monitored eS26 protein levels over time in whole cell extracts after switching the $P_{GAL1}$-*TSR2* strain to repressive glucose containing media. These analyses revealed that eS26 protein levels decreased over time upon Tsr2-depletion (***Figure 5C***).

We observed that purified recombinant eS26 was highly prone to aggregation. Expressing eS26 as a fusion protein with a highly soluble GST tag suppressed its tendency to aggregate. However, removal of the GST tag after cleavage by PreScission protease resulted in immediate aggregation of free eS26, as determined by a massive increase in the light scattering intensity (***Figure 5D***). We tested whether Tsr2 could suppress the aggregating ability of recombinant eS26. We treated GST-eS26 with PreScission protease in absence and presence of Tsr2. A concomitant decrease in the light scattering of the reaction

mixture was observed (*Figure 5D*), as the Tsr2 concentration in the cleavage buffer was increased, indicating aggregation of free eS26 was suppressed. Altogether, these data indicate that Tsr2 protects eS26, and thereby ensures a safe transfer to the 90S pre-ribosome.

## An eS26 mutant associated with Klippel-Feil syndrome in Diamond-Blackfan anemia patients is impaired in importin binding

Mutations in r-proteins have been linked to Diamond-Blackfan anemia (DBA), a rare congenital red blood cell aplasia (*Ellis and Lipton, 2008*; *Ganapathi and Shimamura, 2008*; *Narla and Ebert, 2010*; *Ellis and Gleizes, 2011*; *McCann and Baserga, 2013*; *Ellis, 2014*). Several mutations in the start codon of *RPS26*, including two mutations within eS26, D33N and C77W have been linked to DBA (*Doherty et al., 2010*; *Cmejla et al., 2011*). Both residues are highly conserved from yeast to humans (*Figure 4—figure supplement 3A*, left panel). The C77W mutation is additionally linked to Klippel-Feil syndrome (KFS), a skeletal developmental disorder in DBA patients (*Cmejla et al., 2011*).

In order to analyze the phenotypes induced by the D33N and C77W mutations, we introduced the individual mutations into yeast *RPS26A*. First, plasmids encoding DBA-linked mutants were transformed into the P$_{GAL1}$-*RPS26Arps26b∆* strain and growth was analyzed on glucose containing media. Whereas the D33N mutant partially rescued the lethality of the eS26-conditional mutant, the C77W variant did not allow any growth (*Figure 6A*). Further, as in the P$_{GAL1}$-*TSR2* strain under repressive conditions, both variants resulted in strongly reduced eS26 protein levels (*Figure 6B*). Neither strain displayed defects in the nuclear export of pre-40S subunits (*Figure 1—figure supplement 1B*). As expected, neither variant was able to rescue the 20S pre-rRNA processing defect of eS26-deficient cells, as determined by the strong cytoplasmic localization of Cy3-ITS1 (*Figure 6C*). Thus, eS26 mutants linked to DBA are impaired in cytoplasmic processing of 20S pre-rRNA.

We tested whether the identified eS26 binders could interact with D33N and C77W variants in vitro. Pull-down assays demonstrated that both mutant proteins efficiently bound Tsr2 (*Figure 6D*), suggesting these mutations do not contribute to the Tsr2:eS26 interaction surface. The eS26D33N mutant efficiently binds to Kap123, Kap104 and Pse1 (*Figure 6E*). In agreement with these interaction studies, nuclear uptake of GFP-eS26D33N was not affected (*Figure 6F*) and the levels of GFP-eS26D33N were strongly reduced upon Tsr2-depletion (*Figure 4—figure supplement 2E*). In contrast, the eS26C77W mutant interacted weakly with these importins (*Figure 6E*). We were unable to localize GFP-eS26C77W; whole cells extracts revealed that GFP-eS26C77W protein levels were strongly reduced (*Figure 4—figure supplement 2G*).

## Discussion

A growing yeast cell manufactures ~200,000 ribosomes during one generation time (*Warner, 1999*). This process requires the import of ~14 million r-proteins into the nucleus through ~200 NPCs. Such a process entails rapid transport of importin:r-protein complexes into the nucleus, and necessitates an efficient mechanism to dissociate these complexes to terminate the import process. This permits rapid recycling of importins back to the cytoplasm for subsequent rounds of import. Although it is recognized that r-proteins employ multiple import pathways to reach the nuclear compartment, it remains unclear how these intrinsically unstable and aggregation-prone proteins are targeted to the assembling pre-ribosome. It is assumed that, like a typical import cargo, RanGTP releases the r-protein from the importin and the r-protein somehow finds its way to its cognate rRNA site. Here, we reveal that Tsr2 extracts eS26 from importins and ensures its safe transfer to the 90S pre-ribosome. These data implicate an atypical RanGTP-independent mechanism that terminates the import process, and uncovers an unanticipated link between the nuclear import machinery and the ribosome assembly pathway.

### eS26 is recruited to the 90S pre-ribosome

Using Western analyses and targeted SRM assays, we found that untagged eS26 is recruited to Noc4-TAP and co-enriches with nuclear pre-40S subunits that contain 20S pre-rRNA (*Figure 3A,B*, *Figure 3—figure supplement 1A*). Moreover, eS26-GFP accumulated in the nucleus of *yrb2∆* cells that are specifically impaired in 40S pre-ribosome export (*Figure 3C*). These data suggest that eS26 can be transported to the 90S pre-ribosome. Our findings contrast a previous report wherein a FLAG-tagged eS26 immunoprecipitated mainly 18S rRNA (*Ferreira-Cerca et al., 2007*). In addition, eS26 was not identified in mass spectrometry studies of a pre-40S subunit, and was suggested to replace the assembly factor Pno1 (*Strunk et al., 2011*, *2012*). However, we found that Pno1-TAP efficiently co-enriched eS26

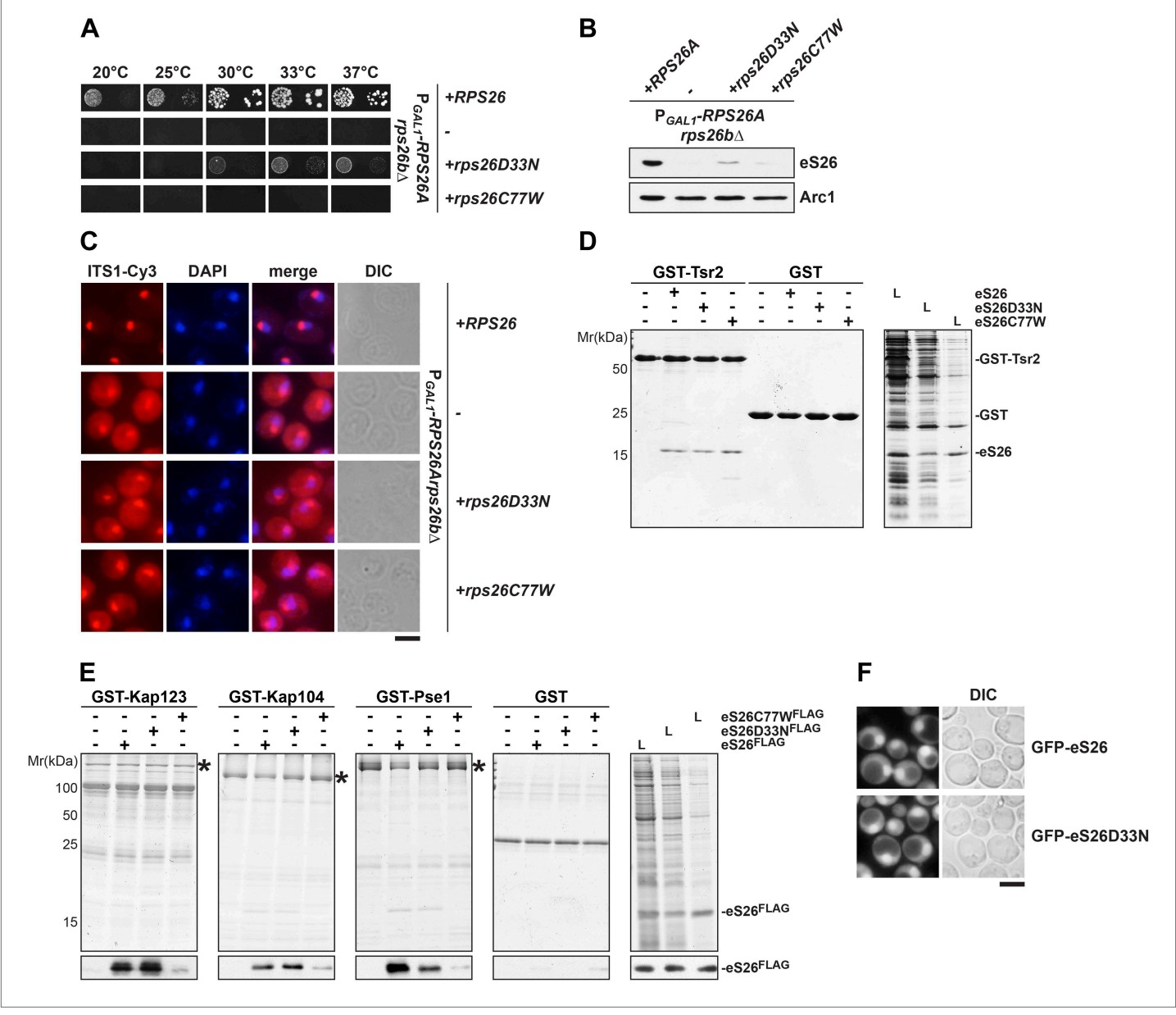

**Figure 6**. The eS26C77W mutant associated with Klippel-Feil syndrome in Diamond-Blackfan anemia patients is impaired in binding importins. (**A**) The DBA linked eS26D33N and eS26C77W mutants are unable to fully rescue the growth defect of eS26-depleted cells. The P$_{GAL1}$-*RPS26Arps26bΔ* strain transformed with different plasmids encoding eS26 mutants were spotted in 10-fold dilutions on selective glucose containing plates and grown at indicated temperatures for 3–7 days. Residues mutated in DBA are depicted in *Figure 4—figure supplement 3A*. (**B**) DBA linked mutations cause strongly reduced eS26 protein levels. Whole cell extracts were prepared from P$_{GAL1}$-*RPS26Arps26bΔ* cells transformed with indicated plasmids encoding for eS26 WT and mutant proteins. eS26 protein levels were assessed by Western analyses using α-eS26 antibodies. Arc1 served as loading control. (**C**) eS26 mutants linked to DBA accumulate 20S pre-rRNA in the cytoplasm. P$_{GAL1}$-*RPS26Arps26bΔ* cells transformed with plasmids encoding for eS26 WT and mutant proteins were grown at 37°C to mid-log phase in glucose containing medium. Localization of 20S pre-rRNA was analyzed by FISH using a Cy3-labeled oligonucleotide complementary to the 5′ portion of ITS1 (red). Nuclear and mitochondrial DNA was stained with DAPI (blue). Scale bar = 5 μm. (**D**) Tsr2 interacts with eS26 mutants linked to DBA. Recombinant GST-Tsr2 was immobilized on Glutathione Sepharose and then incubated with *E. coli* lysates containing eS26a$^{FLAG}$, eS26D33NFLAG or eS26C77WFLAG lysates for 1 hr at 4°C. Bound proteins were eluted by SDS sample buffer, separated by SDS-PAGE and detected by Coomassie Blue staining. L = input. (**E**) eS26C77W is impaired in binding to Kap123, Kap104 and Pse1. Recombinant GST-Kap123, -Kap104, -Pse1 and GST alone were immobilized on Glutathione Sepharose and then incubated with *E. coli* lysate containing eS26$^{FLAG}$, eS26D33NFLAG or eS26C77WFLAG for 1 hr at 4°C. Bound proteins were eluted in SDS sample buffer, separated by SDS-PAGE and visualized by Coomassie Blue staining and Western analyses using α-eS26 antibody. L = input. (**F**) The GFP-eS26D33N fusion protein is efficiently targeted to the nucleus. WT cells expressing GFP-eS26 and GFP-eS26D33N were grown in synthetic media at 30°C to mid-log phase and the localization of GFP-eS26 was analyzed by fluorescence microscopy. Scale bar = 5 μm.

(*Figure 3A*). Further, Tsr2-depletion impaired recruitment of Pno1 to pre-40S subunits, suggesting that eS26 helps to recruit Pno1 (*Figure 5A*). eS26-depletion impaired only 20S pre-rRNA processing in the cytoplasm, suggesting that eS26 does not apparently affect 90S assembly per se, but is specifically required for final maturation (*Figure 2B*). Although the precise timing of eS26 recruitment remains unclear, based on our data, we propose that it is a late event during 90S assembly.

eS26 clamps the 3'-end of mature 18S rRNA (*Figure 4—figure supplement 3A*, right panel; *Rabl et al., 2011*), precisely at the site where the endonuclease Nob1 cleaves the 20S pre-rRNA. Pre-40S subunits that lack eS26 escape nuclear proofreading and are efficiently transported into the cytoplasm. These incompletely assembled pre-40S subunits recruit the endonuclease Nob1 (*Figure 5A*) and form 80S-like particles (*Figures 1D and 2C*). However, they fail to process 20S pre-rRNA (*Figures 1C and 2B*), an essential pre-requisite to form a mature 40S subunit. Thus, 20S pre-rRNA within an 80S-like particle becomes an optimal substrate for Nob1 only when the pre-40S subunit has satisfied a checklist that assesses its potential to translate, including the incorporation of eS26. We propose that the cytoplasmic 20S pre-rRNA cleavage functions as one of the checkpoints that prevent incompletely assembled, pre-40S subunits from entering translation.

## RanGTP-independent dissociation of importin:eS26 complexes by Tsr2

eS26 is targeted to the 90S pre-ribosome and therefore must reach the nucleolus. Unlike the Kap104 adaptor Syo1 that co-imports uL18 (yeast Rpl5) and uL5 (yeast Rpl11) (*Kressler et al., 2012*), Tsr2 does not mediate interactions between eS26 and importins. Instead, our data identified Kap123 and Kap104 as the major importins that directly bind and transport eS26 into the nucleus (*Figure 4*). Recruitment of RanGTP did not efficiently trigger the dissociation of importin:eS26 complexes (*Figure 4*). One possibility could be that eS26 engages in a novel interaction with the importins, thereby delaying its release. Such a delay may ensure the coordinated handover to the next binding factor, Tsr2. Structural analyses of the importin:eS26 complex should provide clues into why eS26 is inefficiently released from importins by RanGTP.

In contrast to RanGTP, Tsr2 efficiently removed eS26 from its importins (*Figure 4*, *Figure 4—figure supplement 3*), identifying an atypical RanGTP-independent mechanism to terminate the import cycle. The observation that Tsr2 prevents proteolysis and aggregation of eS26 (*Figure 5*) indicates an additional 'private' chaperone function. Thus our study adds Tsr2:eS26 to the growing list of known chaperones:r-proteins pairs (Sqt1:uL16; Rrb1:uL3; Yar1:uS3) required for ribosome assembly (*Eisinger et al., 1997*; *Iouk et al., 2001*; *Schaper et al., 2001*; *Koch et al., 2012*). Tsr2 may prevent eS26 from undergoing non-specific interactions with nucleic acids during its journey towards the 90S pre-ribosome. How eS26 is transferred from Tsr2 to 90S pre-ribosomes remains unclear. It is tempting to speculate that posttranslational modifications and/or energy consuming enzymes couple the extraction of eS26 from Tsr2 and subsequent incorporation.

Based on our data, we propose a model in which eS26 is transported to the nuclear compartment predominantly by importins Kap123 and Kap104 (*Figure 7*). Inside the nucleus, eS26 is removed from its importins in a RanGTP-independent mechanism mediated by Tsr2. The released eS26 forms a stable complex with Tsr2. After Tsr2:eS26 complex formation, Tsr2 guarantees a safe transfer of eS26 to the 90S pre-ribosome. Although RanGTP is able to inefficiently release eS26 from its importin, failure to immediately bind Tsr2 results in eS26 degradation. Therefore, in absence of Tsr2, only a smaller fraction of eS26 may reach the 90S pre-ribosome, providing a possible explanation as to why Tsr2-deficient cells are severely impaired in growth but are still viable, although the r-protein eS26 is essential. Notably, human Tsr2 can rescue the severe growth defect of the Tsr2-depleted strain (*Figure 1—figure supplement 1C*), strongly suggesting an evolutionarily conserved role of Tsr2 in 40S assembly.

## Etiology of eS26 mutants linked to Diamond-Blackfan anemia and Klippel-Feil syndrome

Similar to the Tsr2-depletion, both DBA mutants (eS26D33N and eS26C77W) accumulate 20S pre-rRNA in the cytoplasm (*Figure 6C*). The eS26C77W mutant interacted poorly with its import receptors, suggesting that the inability to interact with importins may cause its degradation. Cysteine 77 is one of four conserved cysteines within eS26 that coordinates a $Zn^{2+}$ ion (*Figure 4—figure supplement 3A*, right panel; *Rabl et al., 2011*). Our data raise an intriguing possibility that the NLS within eS26 becomes available to interact with importins only when the $Zn^{2+}$ ion is correctly coordinated. In addition to their transport role, importins may select correctly folded eS26. Notably, the eS26D33N mutant interacted

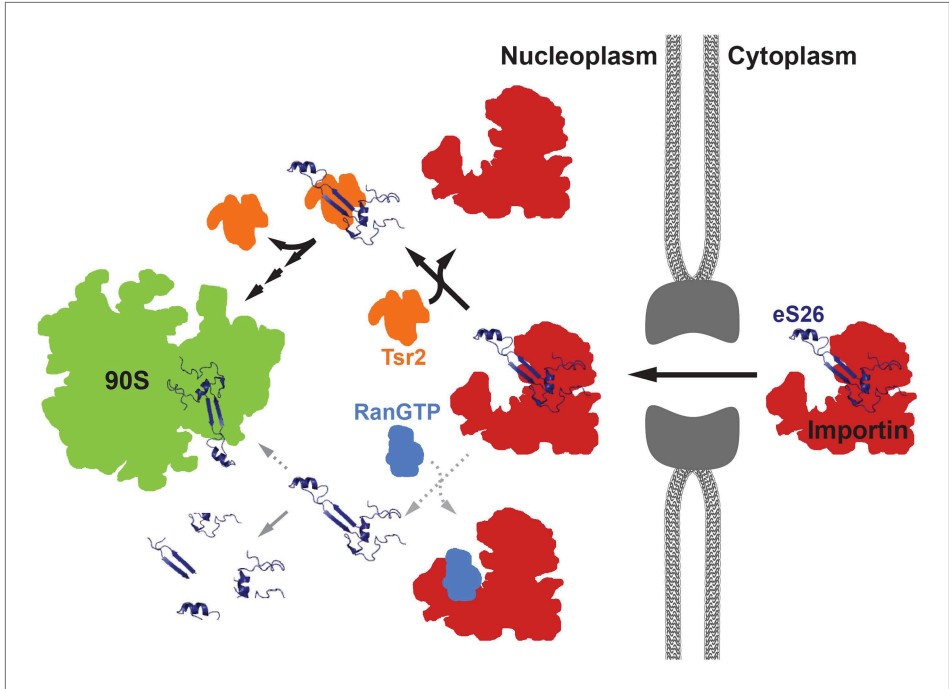

**Figure 7**. A model for the transport of eS26 to the 90S pre-ribosome. Newly synthesized eS26 is transported from the cytoplasm into the nucleus by importins. In the nucleus, Tsr2 alone removes eS26 from importins by a RanGTP-independent mechanism. Subsequently, Tsr2 binds the released eS26, protects it from proteolysis and aggregation, and enables safe transfer to the 90S pre-ribosome. If eS26 is released from the importin by RanGTP it may not immediately encounter Tsr2, resulting in a smaller fraction reaching the 90S pre-ribosome. See 'Discussion' for details of the proposed model.

with Kap123, Kap104, Pse1 and Tsr2 in vitro (***Figure 6D,E***). We speculate that the in vivo instability of this variant might be due to a failure to incorporate eS26 into the 90S pre-ribosome.

Several mutations in eS26 have been linked to DBA, the majority of which are in the start codon, thereby causing eS26 haploinsufficiency (***Doherty et al., 2010***). Notably, eS26 levels are strongly reduced in Tsr2-depleted cells. Interestingly, about half of DBA cases are due to unidentified mutations. Based on these data, we speculate that the *TSR2* gene may be a potential hotspot for DBA.

## A family of escortins?

More than 20 years ago, a system was envisioned to efficiently transfer r-proteins from the NPCs towards the nucleolus (***Russell and Tollervey, 1992***). Here, we identify Tsr2 as the first component of this transfer system that connects the nuclear import machinery with the ribosome assembly pathway. We propose the term 'escortin' to describe this 'linker' function.

Aggregating r-proteins in the nucleolus aggravate the toxicity of a *Caenorhabditis elegans* Huntington disease model and decrease their lifespan ***David et al. (2010)***, emphasizing the importance to safely transfer r-proteins to the assembling pre-ribosomes. Due to their unstable and aggregation-prone nature ***Koplin et al. (2010)*** we envisage an escortin network to securely connect the nuclear import machinery with the ribosome assembly pathway. Intriguingly, like in the case of Kap123:eS26 complex, RanGTP is unable to efficiently release uL14 (human Rpl23a) from importin 7 (RanBP7) (***Jäkel and Görlich, 1998***). Moreover, we found that yeast r-proteins (uS14, eS31a and eS8a) bound to Kap123 were not released upon RanGTP treatment (***Figure 4—figure supplement 4***) suggesting that these r-proteins may require specific escortins for their release. Affinity purifications coupled to mass spectrometry have identified >200 non-ribosomal factors that are directly involved in ribosome assembly (***Bassler et al., 2001***; ***Harnpicharnchai et al., 2001***; ***Dragon et al., 2002***; ***Fatica et al., 2002***; ***Grandi et al., 2002***; ***Nissan et al., 2002***; ***Schäfer et al., 2003***). However, escortins, which are not stably bound to pre-ribosomal particles, may have escaped identification.

Individual subunits/sub-complexes of other macromolecular complexes involved in genome replication, genomic stability and gene expression must be imported into the nucleus prior to their assembly. The fate of these cargoes after being released from importins in the nucleus remains largely unexplored. Many of these components may rely on escortins that will ensure their transfer to their assembly site. Thus, we expect that the list of escortins for ribosome assembly and other biological pathways will expand in the near future.

## Materials and methods

### Yeast strains and plasmids

The *Saccharomyces cerevisiae* strains used in this study are listed in *Supplementary file 1A*. Genomic disruptions, C-terminal tagging and promoter switches at genomic loci were performed as described previously (*Longtine et al., 1998*; *Puig et al., 2001*; *Janke et al., 2004*). Preparation of media, yeast transformations and genetic manipulations were performed according to established procedures.

Plasmids used in this study are listed in *Supplementary file 1B*. Details of plasmid construction will be provided upon request. All recombinant DNA techniques were performed according to established procedures using *E. coli* XL1 blue cells for cloning and plasmid propagation. Point mutations in *RPS26A* were generated using the QuikChange site-directed mutagenesis kit (Agilent Technologies, Santa Clara, CA, USA). All cloned DNA fragments and mutagenized plasmids were verified by sequencing.

### Fluorescence in situ hybridization and microscopy

Localization of 20S pre-rRNA was analyzed using a Cy3-labeled oligonucleotide probe (5′-Cy3-ATG CTC TTG CCA AAA CAA AAA AAT CCA TTT TCA AAA TTA TTA AAT TTC TT-3′) that is complementary to the 5′ portion of ITS1 as previously described (*Faza et al., 2012*).

Pre-40S subunit export, monitored by localization of uS5-GFP and localization of GFP-eS26 was performed as previously described (*Faza et al., 2012*; *Altvater et al., 2014*). Indirect immunofluorescence using affinity-purified polyclonal antibodies against the TAP-tag (1:1000; Thermo Scientific; Rockford, IL, USA) and staining of the nuclear and mitochondrial DNA with DAPI was performed as described previously (*Schlenstedt et al., 1997*; *Solsbacher et al., 1998*).

Cells were visualized using DM6000B microscope (Leica, Germany) equipped with HCX PL Fluotar 63 × /1.25 NA oil immersion objective (Leica, Solms, Germany). Images were acquired with a fitted digital camera (ORCA-ER; Hamamatsu Photonics, Hamamatsu, SZK, Japan) and Openlab software (Perkin–Elmer, Waltham, MA, USA).

### Polysome analyses

Sedimentation analysis of yeast lysates by sucrose gradient ultracentrifugation was performed as described previously (*Kemmler et al., 2009*; *Altvater et al., 2014*). For Western analyses, 500 μl fractions were precipitated by TCA (trichloroacetic acid), washed in acetone, resuspended in 100 μl of onefold SDS sample buffer and separated by SDS-PAGE. Tsr2, eS26 and uL3 were detected by Western analyses. For rRNA analysis, 500 μl fractions were collected and diluted with an equal volume of lysis buffer. RNA was extracted with Phenol-Chlorofom-Isoamylalcohol and precipitated in isopropanol. RNA pellets were washed with 80% ethanol and resuspended in 20 μl water. rRNAs were then separated on a 1.2% Agarose/formaldehyde gel for 1.5 hr at 200 V. For Northern analysis, rRNAs were blotted onto a Hybond-XL (Amersham, Pittsburg, PA, USA) membrane by capillary transfer and probed for 18S (5′-CATGCATGGCTTAATCTTTGAGAC), 20S (5′-GGTTTTAATTGTCCTATAACAAAAGC) and 25S rRNA (5′-TGCCGCTTCACTCGCCGTTAC) using radioactively labeled probes. rRNAs were detected using phosphoimaging screens (GE Healthcare, Pittsburg, PA, USA).

### Tandem affinity purifications (TAPs) and Western analyses

Whole cell extracts were prepared by alkaline lysis of yeast cells as previously described (*Kemmler et al., 2009*).

Tandem affinity purifications (TAP) of pre-ribosomal particles were carried out as previously described (*Faza et al., 2012*; *Altvater et al., 2014*). Calmodulin-eluates were separated on NuPAGE 4–12% Bis-Tris gradient gels (Invitrogen, Carlsbad, CA, USA) and visualized by either Silver staining or Western analyses using indicated antibodies. To analyze RNAs after TAP purification, RNA was extracted with Phenol-Chlorofom-Isoamylalcohol from Calmodulin-eluates and precipitated in isopropanol. RNA

pellets were washed with 80% ethanol and finally resuspended in 20 µl water. 1 µg of total RNA was separated on a 1.2% Agarose/formaldehyde gel for 1.5 hr at 200 V.

Western analyses were performed as previously described (*Kemmler et al., 2009*). The following primary antibodies were used in this study: α-Tsr2/S26 (1:3000; this study), α-Arc1 (1:4000; E Hurt, University of Heidelberg, Heidelberg, Germany), α-uL3 (yeast Rpl3) (1:5000; J Warner, Albert Einstein College of Medicine, Bronx, NY, USA), α-uS7 (yeast Rps5) (1:4000; Proteintech Group Inc., Chicago, IL, USA), α-uS3 (yeast Rps3) (1:3000; M Seedorf, University of Heidelberg, Heidelberg, Germany); α-TAP (CBP) (1:4000; Thermo Scientific, Rockford, IL, USA), α-Pno1 (1:10,000; K Karbstein, Scripps Research Institute, Jupiter, FL, USA), α-Dim1 (1:10,000; K Karbstein, Scripps Research Institute, Jupiter, FL, USA), α-Nob1 (1:500; Proteintech Group Inc., Chicago, IL, USA), α-Tsr1 (1:10,000; K Karbstein, Scripps Research Institute, Jupiter, FL, USA), α-Ltv1 (1:5000; K Karbstein, Scripps Research Institute, Jupiter, FL, USA), α-Rio2 (1:1000; Proteintech Group Inc., Chicago, IL, USA), α-FLAG (1:3000; Sigma-Aldrich, St. Louis, MO, USA). The secondary HRP-conjugated α-rabbit and α-mouse antibodies (Sigma-Aldrich, USA) were used at 1:1000-1:5000 dilutions. Protein signals were visualized using Immun-Star HRP chemiluminescence kit (Bio-Rad Laboratories, Hercules, CA, USA) and captured by Fuji Super RX X-ray films (Fujifilm, Tokyo, Japan).

## Recombinant protein expression and binding assays

All recombinant proteins were expressed in *E. coli* BL21 cells by IPTG induction. $His_6$-tagged proteins were affinity purified in 50 mM Hepes pH 7.5, 50 mM NaCl, 10% glycerol using Ni-NTA Agarose (GE healthcare), GST fusion proteins were purified in PBSKMT (150 mM NaCl, 25 mM sodium phosphate, 3 mM KCl, 1 mM MgCl2, 0.1% Tween, pH 7.3) using Glutathione Sepharose (GE healthcare). GST-tagged importins, $His_6$-taggged importins and RanGTP ($His_6$-Gsp1Q71L-GTP) were expressed and purified as previously described (*Solsbacher et al., 1998*; *Maurer et al., 2001*; *Fries et al., 2007*).

Recombinant GST-Tsr2 was immobilized in PBSKMT on Glutathione Sepharose (GE healthcare), and incubated with *E. coli* lysates containing recombinant eS26, eS26[FLAG], eS26D33NFLAG, eS26C77WFLAG for 1 hr at 4°C. After incubation, the immobilized GST-proteins were washed three times with PBSKMT 4°C. The bound proteins were eluted with LDS. The in vitro binding studies between recombinant eS26[FLAG], eS26D33NFLAG, eS26C77WFLAG, Tsr2, Tsr2:eS26 complex and yeast importins as GST-fusion proteins were performed as previously described (*Solsbacher et al., 1998*). 1/5th of the bound proteins and input (eS26, eS26[FLAG], eS26D33NFLAG, eS26C77WFLAG) were analyzed on a Coomassie Blue stained gel. 1/10th of the bound proteins and 1/1000th of the input was used for Western analyses.

To dissociate the GST-importin:eS26[FLAG] (Kap123, Pse1 and Kap104) complex or GST-Kap123:eS31[FLAG], GST-Kap123:eS8[FLAG], GST-Kap123:eS14[FLAG] complexes pre-immobilized GST-importin:ribosomal protein complexes were incubated with buffer alone or 3 nM of 3'-end of 18S rRNA (only for eS26[FLAG]), 1.5 µM Tsr2, 1.5 µM $His_6$-Tsr2 (only eS26[FLAG]) and/or 1.5 µM RanGTP ($His_6$-Gsp1Q71L-GTP) for 1 hr at 4°C (protocol modified from *Rothenbusch et al., 2012*). To show that eS26 stably associated with Tsr2 after release from importins, the supernatant of the samples with buffer alone and $His_6$-Tsr2 were incubated with Ni-NTA Agarose for 1 hr at 4°C. For dissociation kinetics, 1.5 µM RanGTP ($His_6$-Gsp1Q71L-GTP) or Tsr2 were added to pre-immobilized importin:eS26[FLAG] complexes and samples were withdrawn at 1, 2, 4 and 8 min. Bound proteins were eluted in twofold LDS/SDS-sample buffer by incubating at 70–95°C and separated by SDS-PAGE. Proteins were visualized by Coomassie Blue staining or by Western analyses using antibodies against Tsr2 and eS26.

## Aggregation assay

The aggregation assay was performed in a 384-well plate (Polystyrene, clear bottom, low volume, Corning, USA). In each well 33 µM GST-eS26 and a given concentration of Tsr2 (0 up to 266 µM) in PBSKMT was pre-incubated for 1 hr at 4°C (final volume: 90 µl). 250 nM of PreScission protease was added to initiate aggregation. Aggregation of free eS26 was measured at 450 nm using a Multiskan GO plate reader (Thermo Scientific, USA). As controls, scattering intensities of individual components used in the aggregation assay such as 33 µM of GST-eS26 alone, 266 µM of Tsr2 alone, PreScission protease and buffer were measured. Four replicates were performed for each sample measured.

## SRM assay development, quantitation and statistical analysis

### Sample preparation

Affinity purified pre-40S particles were processed for mass spectrometric analysis as described earlier (*Altvater et al., 2012*). Affinity-purified protein samples were denatured and cysteine residues were reduced and alkylated. After tryptic digest the peptides were purified with C18 columns. Before mass

spectrometric analysis, 11 retention time calibration peptides (iRT peptides, RT-kit WR, Biognosys) were added to every sample at a ratio of 1:20.

## SRM assay development

To develop SRM assays, peptide samples of the affinity purified pre-40S particles were analyzed on a nanoLC 1Dplus system (Eksigent) connected to a TripleTOF 5600+ mass spectrometer (ABSciex). Peptides were separated by reversed-phase liquid chromatography on a 20-cm fused silica microcapillary (75 µm inner diameter, New Objective) packed in-house with 3 µm C18 beads (Magic C18 AQ, 200 Å pore size; Michrom BioResources, Auburn, CA, USA) with a linear gradient from 98% solvent A (98% acetonitrile, 0.1% formic) and 2% solvent B (98% acetonitrile, 0.1% formic acid) to 35% solvent B over 120 min at a flow rate of 300 nl/min. The mass spectrometer was operated in information-dependent acquisition (IDA) mode. MS1 spectra were recorded in the range of 360–1460 m/z for 500 ms. Up to 20 precursor ions with charge state 2–5 were selected for fragmentation and MS2 spectra were recorded in the range of 50–2000 m/z for 150 ms in high sensitivity mode. Selected precursor ions were excluded for 20 s after one occurrence. Raw data files were centroided and converted to mzML format using the ABSciex Data Converter and then converted to mzXML format using ProteoWizard MSConvert (*Kessner et al., 2008*).

MS2 spectra were searched with Sorcerer-SEQUEST (SageN Research) against a *S. cerevisiae* protein database (SGD, May 2013) to which the sequences of the 11 spiked-in iRT peptides and various common contaminants were added. Reversed sequences of all proteins were appended to the protein database to assess the number of false positive peptide-spectrum matches (*Elias and Gygi, 2007*). Tryptic cleavage was defined to occur after lysine and arginine, unless followed by a proline residue, and peptides were allowed to have up to one non-tryptic end and up to two missed cleavages. Cysteine carbamidomethylation was added as static modification and methionine oxidation as variable modification. Precursor mass tolerance was set to 50 ppm. Resulting peptide-spectrum matches were statistically assessed using PeptideProphet and iProphet as part of the TPP (*Keller et al., 2002*; *Deutsch et al., 2010*; *Shteynberg et al., 2011*). The iProphet output was processed with MAYU (*Reiter et al., 2009*), which has been modified to work with iProphet probabilities. Peptide-spectrum matches were selected at a false discovery rate (FDR) of 0.07% to obtain a protein FDR of 1%. An in-house written script was used to convert all retention times into iRT values (*Escher et al., 2012*). SpectraST (*Lam et al., 2008*) was used to generate a consensus spectral library from which the six most intense fragment ions (b- or y-ions) per peptide precursor were selected in Skyline (*MacLean et al., 2010*). The final SRM assays for target proteins and iRT peptides are given in *Supplementary file 2*.

## SRM analysis

The SRM data was acquired on a TSQ Vantage triple quadrupole mass spectrometer (Thermo Fisher Scientific) coupled to a nanoLC 1Dplus system (Eksigent). Peptides were separated by reversed-phase liquid chromatography on a 10.5-cm fused silica microcapillary (75 µm inner diameter, New Objective) packed in-house with 5 µm C18 beads (Magic C18 AQ, 200 Å pore size; Michrom BioResources) with a linear gradient from 95% solvent A (98% acetonitrile, 0.1% formic) and 5% solvent B (98% acetonitrile, 0.1% formic acid) to 35% solvent B over 35 min at a flow rate of 300 nl/min. The mass spectrometer was operated in positive mode using electrospray ionisation with a voltage of 1400 V. The capillary temperature was set to 280°C and the collision gas pressure to 1.5 mTorr. All transitions were monitored in scheduled mode with a retention time window of ±600 s, a cycle time of 2 s, and a mass window of 0.7 of half-maximum peak width (unit resolution) in Q1 and Q3.

The SRM data was analyzed manually in Skyline (*MacLean et al., 2010*). After removing non-detectable peptides and interfered transitions, peptide intensities (sum of integrated transition peak area) were exported for further processing in Excel. All peptides were normalized on the protein abundance of uS7 (Rps5). Peptides of each protein were ranked by their average intensity over all samples and the three most intense peptides were averaged to obtain an abundance value for every protein. The standard deviation was calculated assuming that the values are a sample of the entire population.

The SRM data can be viewed in and downloaded from Panorama: https://daily.panoramaweb.org/labkey/project/Aebersold/schubert/2014_Schuetz_Ribo-40S/begin.view.

## Acknowledgements

We are grateful to M Peter, E Michel, G Braus, K Karbstein, M Seedorf for generously sharing plasmids, strains and antibodies. We thank J Ottl, O Esser, D Barlier and R Brunner from the Novartis Institute of

Biomedical research, Basel for protein aggregation measurements. We thank S Guthörl for technical assistance. We thank R Aebersold and J Pfannstiel for mass spectrometry, J Matos, C Azzalin, all members of the Panse laboratory and in particular C Weirich for enthusiastic discussions.

## Additional information

### Funding

| Funder | Grant reference number | Author |
|---|---|---|
| Swiss National Science Foundation | | Vikram G Panse |
| Swiss Federal Institute of Technology Zurich | | Vikram G Panse |
| European Research Council | Starting Grant EURIBIO260676 | Vikram G Panse |
| Deutsche Forschungsgemeinschaft | | Gabriel Schlenstedt |
| Universität des Saarlandes | Homburger Forschungsförderung | Gabriel Schlenstedt |
| European Research Council | Advanced Grant 233226 | Olga T Schubert |
| European Union | PRIME-XS 262067 | Olga T Schubert |

The funders had no role in study design, data collection and interpretation, or the decision to submit the work for publication.

### Author contributions

SS, UF, MA, Conception and design, Acquisition of data, Analysis and interpretation of data, Drafting or revising the article; PN, MG, SC, Acquisition of data, Analysis and interpretation of data, Contributed unpublished essential data or reagents; CP, YC, OTS, Conception and design, Acquisition of data, Analysis and interpretation of data; GS, Conception and design, Analysis and interpretation of data; VGP, Conception and design, Analysis and interpretation of data, Drafting or revising the article

## Additional files

### Supplementary files

• Supplementary file 1. (**A**) Yeast strains used in this study. (**B**) Plasmids used in this study.

• Supplementary file 2. Peptides used for SRM assays.

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
