## [Decision Letter]

Thank you for sending your work entitled “A RanGTP-independent Mechanism Allows Ribosomal Protein Nuclear Import for Ribosome Assembly” for consideration at *eLife*. Your article has been favorably evaluated by Randy Schekman (Senior editor) and 2 reviewers, one of whom is a member of our Board of Reviewing Editors.

The Reviewing editor and the other reviewers discussed their comments before we reached this decision, and the Reviewing editor has assembled the following comments to help you prepare a revised submission.

This manuscript describes the function of the ribosome assembly factor Tsr2 as a chaperone for the essential ribosomal protein eS26 in yeast. Originally identified in a RNA processing screen as needed for 20S rRNA processing, the function of Tsr2 was poorly understood. Through a series of systematic analyses, the authors trace its role to pre-40S maturation and subsequently to ribosomal protein eS26 incorporation. Biochemical experiments illustrated a direct physical interaction between Tsr2 and eS26 that was mutually exclusive of the eS26-importin interaction. Further analysis showed that Tsr2, and not Ran-GTP, efficiently disassembles the eS26-importin complex in a substrate-specific reaction. These findings, together with the observation that eS26 is both aggregation prone and degradation prone in the absence of Tsr2, lead the authors to an interesting model whereby Tsr2 represents a new functional family of proteins that mediate the exchange of ribosomal proteins from importins to the preribosome.

The referees agreed that the data are of very high technical quality, represent a large body of work, and are nicely organized and clearly described in the paper. The primary criticism centers around the issue of solidifying the central conclusion that Tsr2 is localized to and acting in the nucleus after eS26 import. Since earlier studies have suggested that eS26 is loaded onto the preribosome in the cytosol, it is worth more rigorously excluding this possibility. In this alternative interpretation of the authors' data, Tsr2 would bind to eS26 in the cytoplasm to prevent its interaction with karyopherins and to promote its late assembly, possibly in the cytoplasm. Addressing the following points would help discriminate these models.

1) Tsr2 has previously been reported to be both nuclear and cytoplasmic. The authors monitored the localization of Tsr2 coupled to 3 copies of GFP and expressed on top of WT protein. The higher expression level due to the extra copies from the CEN vector could lead to a mislocalization artifact. The authors should nail down the localization of functional/endogenous Tsr2 by either using an integrated GFP-tagged functional protein or indirect IF using anti-Tsr2. This would strengthen their model regarding its site of action.

2) It is unclear in Figure 4 why GFP-eS26 accumulates in the nucleus in WT cells. It should be incorporated into ribosomes and exported. Its accumulation in the nucleus suggests that it may not load efficiently into ribosomes. This is an important point because if the protein is not functional and not incorporated into ribosomes, its localization may be aberrant. The authors should clarify whether GFP-eS26 is functional and incorporated into ribosomes.

3) If the argument is that free eS26 is unstable when not bound by TSR2 or Kaps, then why do we apparently see a strong eGFP-eS26 signal in the cytoplasm in the absence of Kap123 (assuming identical imaging conditions for all panels). Presumably this is protein that is not bound by a Kap, hence its cytoplasmic localization, and not bound by Tsr2, which should be restricted to the nucleus. The apparent stability seems counter to the model. Again, the knowing the functionality of eGFP-eS26 would help with interpreting this result.

4) If the D33 and C77 mutant eS26 proteins cannot interact with Kap123, they should remain in the cytoplasm. This should be investigated to correlate a loss of binding in vitro with an in vivo effect.

Minor comments:

1) The title is not a very good description of what is actually discovered. First, it's too general given that for now, the Tsr2-eS26 system is unique. Second, it implies that Tsr2 allows nuclear import, but actually, Tsr2 acts after the import has happened. I therefore suggest something more specific such as “Tsr2 captures a ribosomal protein from its nuclear importin to facilitate incorporation into an assembling pre-ribosome”.

2) The data are all quite clear in general, but I was slightly confused by what I'm looking at with the stained gels of the GST pulldowns. In most cases, the GST tagged protein is a minor species. Can the authors comment on what the other bands are (*E. coli* lysate proteins, impurities of the GST-purified protein, something else)?

3) Please indicate what the numbers to the side of the images in Figure 4 mean. Presumably a quantitative assessment of nuclear localization?

4) The authors may want to explain in one or two sentences what SRM is for the unfamiliar reader.

5) It appears to my eye in Figure 4—figure supplement 2 that 375 nM RanGTP binds importin and releases substrate better than 1.5 uM. Are the panels reversed, or are these differences not significant?

6) To rule out effects of Kap123 on Tsr2, the authors should report the localization of Tsr2 in kap123Δ cells.

7) Is it possible that eS26 interaction with Kaps is nonspecific and electrostatic, such that it can be competed off by Tsr2, a true binding partner?

8) Figure 6. Why do the *E. coli* lysates expressing WT and mutant eS26 look so different? Is the C77 mutant less soluble?

9) The authors should cite other examples of chaperones – Yar1, for uS3 and Sqt1 for uL16.

10) Figure 1. I don't think it can be 100% of input loaded on the gel.

11) Figure 3. The data would be more convincing if one of the proteins had shown differential binding. Possibly uS3.

---

## [Author Response]

We have performed additional experiments and clarified the issues raised by the reviewers to strengthen the conclusion that Tsr2 localizes to the nucleus and functions in targeting eS26 to the 90S pre-ribosome.

*The referees agreed that the data are of very high technical quality, represent a large body of work, and are nicely organized and clearly described in the paper. The primary criticism centers around the issue of solidifying the central conclusion that Tsr2 is localized to and acting in the nucleus after eS26 import. Since earlier studies have suggested that eS26 is loaded onto the preribosome in the cytosol, it is worth more rigorously excluding this possibility. In this alternative interpretation of the authors' data, Tsr2 would bind to eS26 in the cytoplasm to prevent its interaction with karyopherins and to promote its late assembly, possibly in the cytoplasm. Addressing the following points would help discriminate these models*.

A) Where is endogenous Tsr2 localized?

We have localized endogenous Tsr2 using an integrated -TAP and -GFP tag at the genomic locus. These studies show that Tsr2-TAP and Tsr2-GFP predominantly localize to the nucleus (Figure 1).

B) Is eS26 targeted to the 90S pre-ribosome?

Tsr2-TAP co-enriches stoichiometric amounts of eS26 (Figure 1). Further, Western analyses using antibodies directed against eS26 and selected reaction monitoring mass spectrometry (SRM-MS) showed that eS26 co-enriches with multiple nucleolar/nuclear pre-ribosomal particles in the 40S maturation pathway (Figure 3 and Figure 3). These data strongly suggest that eS26 can be targeted to the nucleus for incorporation into the 90S pre-ribosome.

To strengthen the biochemical data above, we have performed an additional experiment. The ribosomal protein uS5 is transported to the nucleus and then assembled into the 90S pre-ribosome, the precursor for the 40S pre-ribosome. If nuclear export of the 40S pre-ribosome is impaired, uS5 accumulates in the nucleus: for e.g. in the *yrb2Δ* mutant, uS5-GFP accumulates in the nucleus (Figure 1—figure supplement 1). Likewise, if eS26 were loaded on the 90S pre-ribosome in the nucleus, then it should also accumulate in the nucleus in the *yrb2Δ* strain. To test this, we monitored localization of C-terminally GFP tagged eS26 (eS26-GFP) in WT and *yrb2Δ* cells. In WT cells, eS26-GFP localizes to the cytoplasm. In agreement with the nuclear loading model, the *yrb2Δ* mutant accumulates eS26-GFP in the nucleus (Figure 3). These cell-biological data support the idea that eS26-GFP is targeted to the 90S pre-ribosome. We have incorporated these findings in the Results Section and included these cell-biological data in Figure 3.

*1) Tsr2 has previously been reported to be both nuclear and cytoplasmic. The authors monitored the localization of Tsr2 coupled to 3 copies of GFP and expressed on top of WT protein. The higher expression level due to the extra copies from the CEN vector could lead to a mislocalization artifact. The authors should nail down the localization of functional/endogenous Tsr2 by either using an integrated GFP-tagged functional protein or indirect IF using anti-Tsr2. This would strengthen their model regarding its site of action*.

Is the nuclear location of the Tsr2-3xGFP fusion in WT cells an artifact? The following observations argue against this possibility:

We have localized endogenous Tsr2 using an integrated -TAP and -GFP tag by immunofluorescence. Tsr2-TAP and Tsr2-GFP localize predominantly to the nucleus as determined by the co-localization of the Alexa Fluor 568 (red) and DAPI (blue) signals and GFP fluorescence (Figure 1). Tsr2-TAP and Tsr2-GFP strains are not impaired in growth like the Tsr2-depleted strain suggesting that the fusion proteins are functional (Figure 1). The Tsr2-3xGFP produced from a CEN plasmid (under its natural promoter and terminator) also localizes predominantly to the nucleus in Tsr2-depleted cells (Figure 1) and rescues the severe growth phenotype of a Tsr2-depleted strain indicating it is functional (Figure 1).

*2) It is unclear in*
Figure 4
*why GFP-eS26 accumulates in the nucleus in WT cells. It should be incorporated into ribosomes and exported. Its accumulation in the nucleus suggests that it may not load efficiently into ribosomes. This is an important point because if the protein is not functional and not incorporated into ribosomes, its localization may be aberrant. The authors should clarify whether GFP-eS26 is functional and incorporated into ribosomes*.

Different lines of evidence (stated above) support the idea that eS26 is targeted to the 90S pre-ribosome. *In vitro* binding studies revealed that amongst the 11 yeast importins Kap104, Kap123 and Pse1 directly interacted with eS26. To verify these interaction data *in vivo* we have used the GFP-eS26 fusion protein as a tool to monitor the nuclear targeting of eS26. We apologize to the reviewers for not explaining the use of this fusion protein.

It is complicated to monitor the nuclear uptake of eS26 *in vivo* (and other ribosomal proteins) since it finally resides in the cytoplasm. One strategy is to uncouple nuclear import of eS26 from its export. This would mean: to allow eS26 import, but selectively impair eS26 recruitment to the 90S pre-ribosome. We took advantage of the crystal structure of 40S subunit ([73], doi: 10.1126/science.1198308). In contrast to the C-terminus, the N-terminus of eS26 is deeply embedded within the 18S rRNA framework (Figure 4—figure supplement 2). We therefore fused GFP to the N-terminus of eS26, with the aim of constructing an eS26 that fails to be incorporated into the 90S pre-ribosome. Sucrose gradient analyses showed that GFP-eS26 co-sediments in lighter fractions at the top of the gradient suggesting that it is not incorporated into pre-ribosomes (Figure 4—figure supplement 2). Binding studies showed that like eS26, GFP-eS26 interacts with Kap123 and Kap104 (Figure 4—figure supplement 2). Thus, like eS26, GFP-eS26 is functional to recruit specific components of the import machinery. Consistent with these studies, GFP-eS26 is efficiently targeted to the nucleus in a Kap123 and Kap104 dependent manner. Therefore, we do not think the nuclear location of GFP-eS26 is aberrant.

Further, GFP-eS26 directly binds Tsr2 (Figure 4—figure supplement 2) and, like eS26, the stability of the GFP-eS26 depends on Tsr2 (Figure 4—figure supplement 2), i.e. GFP-eS26 is degraded upon Tsr2-depletion. Although the GFP-eS26 fusion does not complement the eS26-depleted strain (Figure 4—figure supplement 2), in combination with the biochemical data it serves as a valuable tool to investigate the nuclear uptake of eS26. We have now explained the use of GFP-eS26 fusion protein to monitor its nuclear import and have included these additional data in Figure 4—figure supplement 2.

*3) If the argument is that free eS26 is unstable when not bound by TSR2 or Kaps, then why do we apparently see a strong eGFP-eS26 signal in the cytoplasm in the absence of Kap123 (assuming identical imaging conditions for all panels). Presumably this is protein that is not bound by a Kap, hence its cytoplasmic localization, and not bound by Tsr2, which should be restricted to the nucleus. The apparent stability seems counter to the model. Again, the knowing the functionality of eGFP-eS26 would help with interpreting this result*.

We have exploited the GFP-eS26 fusion as a proxy to investigate transport of eS26 to the nucleus. Consistent with our binding data, nuclear uptake of GFP-eS26 is impaired in the *kap123Δ* and *kap104Δ* mutant. Western analyses show that the eS26 is stable in the *kap123Δ* and *kap104Δ* mutant (Figure 8).Author response image 1.

It is unlikely that TSR2 and importins are the only factors that protect eS26 *in vivo* during its journey towards the 90S pre-ribosome. It appears that newly synthesized eS26 after emerging from the translating ribosome is stabilized by yet unknown cytoplasmic factors before being captured by importins. Intriguingly, yeast deficient for the ribosome associated NAC and SSB-RAC chaperone systems that associate with newly synthesized polypeptides accumulate ribosomal protein aggregates (including eS26) ([47], doi: 10.1083/jcb.200910074). We speculate that the NAC and SSB-RAC chaperone systems hold eS26 in the cytoplasm before transport to the nucleus.

*4) If the D33 and C77 mutant eS26 proteins cannot interact with Kap123, they should remain in the cytoplasm. This should be investigated to correlate a loss of binding in vitro with an in vivo effect*.

The reviewers appear to have inadvertently mistaken our binding analyses for eS26-mutants linked to DBA. The eS26D33N mutant is not impaired in binding importins and Tsr2 (Figure 6). As expected, nuclear uptake of GFP-eS26D33N is not impaired (Figure 6). Notably, like GFP-eS26, the *in vivo* stability of the nuclear-targeted GFP-eS26D33N also depends on Tsr2 (Figure 4—figure supplement 2).

In contrast, the eS26C77W mutant weakly interacts with importins *in vitro* (Figure 6). However, we found that the GFP-eS26C77W is unstable in WT cells (Figure 4—figure supplement 2), therefore we were unable to localize GFP-eS26C77W mutant *in vivo*. The inability of eS26 to coordinate the Zn^2+^ ion might trigger cytoplasmic degradation of the GFP-eS26C77W fusion protein. As a complementary experiment, we investigated which region within eS26 contributes to its nuclear uptake. For this, we monitored the location of several truncations of eS26 fused to GFP. These studies suggest that the Zn^2+^-binding domain is required for efficient nuclear uptake of eS26 (Figure 4—figure supplement 2).

Minor comments:

*1) The title is not a very good description of what is actually discovered. First, it's too general given that for now, the Tsr2-eS26 system is unique. Second, it implies that Tsr2 allows nuclear import, but actually, Tsr2 acts after the import has happened. I therefore suggest something more specific such as “Tsr2 captures a ribosomal protein from its nuclear importin to facilitate incorporation into an assembling pre-ribosome”*.

Import cargos are ferried to the nucleus by import receptors. Nuclear import of a cargo is completed only after dissociation from the receptor. For typical import cargos, this irreversible step occurs upon binding of RanGTP to the importin:cargo complex in the nucleus. Here, *in vitro* studies revealed that Tsr2, without RanGTP, is able to dissociate eS26 from an importin:eS26 complex to terminate the import process. Therefore, we suggest Tsr2 “allows” RanGTP-independent nuclear import.

While, we agree with the reviewer that the Tsr2-eS26 system is unique for now, the finding that three other Kap123:ribosomal protein complexes (Kap123:uS14, Kap123:eS31 and Kap123:eS8) cannot be dissociated by RanGTP (Figure 4—figure supplement 4), implicate additional escortins that target these ribosomal proteins to the 90S pre-ribosome. Therefore, we feel that the title reflects the overall findings of the paper.

*2) The data are all quite clear in general, but I was slightly confused by what I'm looking at with the stained gels of the GST pulldowns. In most cases, the GST tagged protein is a minor species. Can the authors comment on what the other bands are (*E. coli *lysate proteins, impurities of the GST-purified protein, something else)*?

Importins are very large proteins (> 120 kDa) and are quite sensitive to proteolysis during purification and/or binding assays. The additional bands in the Commassie stained gel are C-terminal truncated forms of importins fused to a N-terminal GST tag as judged by a Western against GST (see Figure 9, right).Author response image 2.

*3) Please indicate what the numbers to the side of the images in*
Figure 4
*mean. Presumably a quantitative assessment of nuclear localization*?

Indeed, this is a quantitative assessment of the nuclear localization of GFP-eS26. This is now mentioned in the legend of Figure 4.

*4) The authors may want to explain in one or two sentences what SRM is for the unfamiliar reader*.

We have explained SRM and its specific use for detecting and quantitating proteins in the text.

*5) It appears to my eye in*
Figure 4—figure supplement 2
*that 375 nM RanGTP binds importin and releases substrate better than 1.5 uM. Are the panels reversed, or are these differences not significant*?

We have repeated these experiments. As shown in Figure 4—figure supplement 3, there are no significant differences in recruitment of RanGTP to importins at the different concentrations.

*6) To rule out effects of Kap123 on Tsr2, the authors should report the localization of Tsr2 in kap123Δ cells*.

Tsr2-3xGFP mislocalizes in *kap123Δ* cells to the cytoplasm, but not in the *pse1-1* or the *kap104Δ* mutant (Figure 4). Thus, Kap123 seems to be an import receptor for Tsr2. However, we did not observe a direct interaction between Tsr2 and Kap123 or any other importin *in vitro* (Figure 4 and Figure 4—figure supplement 1). One possibility could be that import of Tsr2 by Kap123 is regulated by posttranslational modification. Alternatively, Tsr2 may be transported into the nucleus *via* a “piggy bag” mechanism bound to another yet unknown Kap123 cargo. We can exclude the possibility that eS26 serves as an adaptor to import Tsr2 since (1) Tsr2-3xGFP does not mislocalize to the cytoplasm in a eS26-depleted strain (Figure 4) and (2) *in vitro* binding assays show that the Tsr2:eS26 complex does not interact with Kap123 (Figure 4). These data have been included in Results Section.

*7) Is it possible that eS26 interaction with Kaps is nonspecific and electrostatic, such that it can be competed off by Tsr2, a true binding partner*?

We do not think that the interaction between the importin and eS26 is a non-specific electrostatic interaction for the following reasons: First, all the binding assays have been performed in presence of competing *E. coli* lysates. Second, only 3 of the 11 tested yeast importins, efficiently bind eS26. Cell-biological studies support the idea that Kap104 and Kap123 function as the major importins for eS26. Finally, a single amino acid change in eS26 (eS26C77W) impairs binding to the importins.

*8)*
Figure 6*. Why do the* E. coli *lysates expressing WT and mutant eS26 look so different? Is the C77 mutant less soluble*?

We have used *E. coli* lysates that contain equal amounts of eS26^FLAG^ protein in our binding assays. We have repeatedly observed that the eS26C77W mutant (for unknown reasons) is better expressed as compared to the WT and the eS26D33N mutant. For our binding assays, we need to use much lower amounts of lysate that contains the eS26C77W mutant. This is why the load lysates look different. Please compare Coomassie stained gel containing equal amounts of *E. coli* lysate expressing WT eS26 and mutant proteins (see Figure 10).Author response image 3.

*9) The authors should cite other examples of chaperones – Yar1, for uS3 and Sqt1 for uL16*.

The chaperones Yar1 and Sqt1 are now cited.

*10)*
Figure 1*. I don't think it can be 100% of input loaded on the gel*.

Indeed, it not that we have not loaded the entire (100%) of input. We have now clearly explained the load for our binding assay in the Materials and methods .

*11)*
Figure 3*. The data would be more convincing if one of the proteins had shown differential binding. Possibly uS3*.

We were unable to develop an SRM assay for uS3, since the Triple-Quadruple mass spectrometer in our proteomics facility is booked out until the middle of September. To directly address the concern regarding differential binding, we have performed Western analyses using antibodies directed against uS3 (Figure 3). These studies revealed that uS3 co-enriches with Enp1-TAP but is not found on the earliest 90S pre-ribosome (Noc4-TAP) (Figure 3).